# Thinker: Learning to Plan and Act

**Stephen Chung**
University of Cambridge
mhc48@cam.ac.uk

**Ivan Anokhin**
Mila, Université de Montréal
ivan.anokhin@mila.quebec

**David Krueger**
University of Cambridge
dsk30@cam.ac.uk

## Abstract

We propose the Thinker algorithm, a novel approach that enables reinforcement learning agents to autonomously interact with and utilize a learned world model. The Thinker algorithm wraps the environment with a world model and introduces new actions designed for interacting with the world model. These model-interaction actions enable agents to perform planning by proposing alternative plans to the world model before selecting a final action to execute in the environment. This approach eliminates the need for handcrafted planning algorithms by enabling the agent to learn how to plan autonomously and allows for easy interpretation of the agent's plan with visualization. We demonstrate the algorithm's effectiveness through experimental results in the game of Sokoban and the Atari 2600 benchmark, where the Thinker algorithm achieves state-of-the-art performance and competitive results, respectively. Visualizations of agents trained with the Thinker algorithm demonstrate that they have learned to plan effectively with the world model to select better actions. Thinker is the first work showing that an RL agent can learn to plan with a learned world model in complex environments.

## 1 Introduction

Model-based reinforcement learning (RL) has significantly enhanced sample efficiency and performance by employing world models, or simply models, to generate additional training data [1, 2], provide better estimates of the gradient [3, 4], and facilitate planning [5]. Here, *planning* refers to the process of interacting with the world model to inform the subsequent selection of actions. Handcrafted planning algorithms, such as Monte Carlo Tree Search (MCTS) [6], have achieved remarkable success in recent years [7, 8, 5], underscoring the importance of planning.

However, a significant research gap persists in creating methods that enable an RL agent to *learn to plan* [9], meaning to interact autonomously with a model, eliminating the need for handcrafted planning algorithms. We suggest this shortfall stems from the inherent complexities of mastering several skills concurrently required by planning: searching (exploring new potential plans), evaluating (assessing plan quality), summarizing (contrasting different plans), and executing (implementing the optimal plan). While a handcrafted planning algorithm can handle these tasks, they pose a significant learning challenge for an RL agent that learns to plan. Introducing a learned world model further complicates the issue, as predictions from the model might not be accurate.

To address these challenges, we introduce the *Thinker* algorithm, a novel approach that enables the agent to interact with a learned model as an integral part of the environment. The Thinker algorithm wraps a Markov Decision Process (MDP) with a learned model and introduces a new set of actions that enable agents to interact with the model.[1] The manner in which an agent uses a model is not predetermined. In principle, an agent can learn various common planning algorithms, such as $n$-step exhaustive search or MCTS, or ignore the model if planning is not beneficial. Importantly, we

---

[1]Full code is available at `https://github.com/stephen-chung-mh/thinker`, which allows for using the Thinker-augmented MDP with the same interface as OpenAI Gym [10].

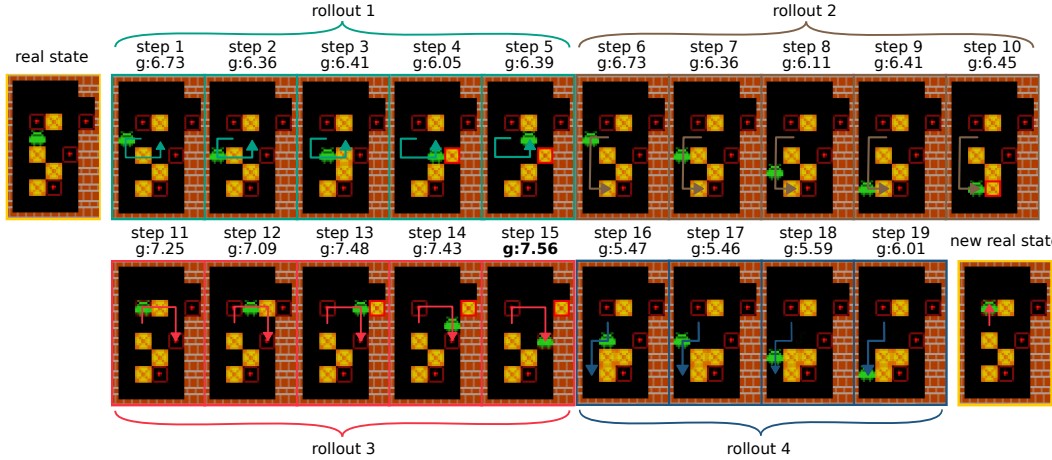

Figure 1: Illustration of a trained agent during a single stage in Sokoban, where the goal is to push all four boxes to red-bordered squares. From the root state, the agent generates four rollouts using the learned model. Except for the two real states, all frames are generated by the model, showcasing near-perfect frame prediction. Predicted rollout return $g$ (equivalent to $g_{root,cur}$ in Equation 1) is displayed. The four rollouts are reasonable plans with agents pushing different boxes towards targets. The agent follows the third rollout for the next five real steps (not fully shown in the figure), likely due to its high rollout return. Crucially, plan generation, plan execution, and the model are all *learned*. Videos of the whole game and other Atari games are available at `https://youtu.be/OAfZh5SR7Fk`.

have designed the augmented MDP in a way that simplifies the processes of searching, evaluating, summarizing, and executing, making it easier for agents to learn to use the model.

Drawing from neuroscience, our work is inspired by the influential hypothesis that the brain conducts planning via *internalized actions*—actions processed within our experience-based internal world model rather than externally [11, 12]. According to this hypothesis, both model-free and model-based behaviors utilize a similar neural mechanism. The main distinction lies in whether actions are directed toward the external world or the internal world model. This perspective is consistent with the structural similarities observed in brain regions governing model-free and model-based behaviors [13]. This hypothesis offers insights into how an RL agent might *learn to plan*—by unifying both imaginary and real actions and leveraging existing RL algorithms.

Experimental results show that actor-critic algorithms, when applied to the Thinker-augmented MDP, yield state-of-the-art performance in Sokoban. Notably, this approach attains a solving rate of 94.5% within 5e7 frames, a significant improvement over the 56.7% solving rate achieved when the same algorithm is applied to the raw MDP. On the Atari 2600 benchmark, actor-critic algorithms using the Thinker-augmented MDP show a significant improvement over those using the raw MDP. Visualization results, as shown in Fig 1, reveal the agent's effective use of the model. To summarize, the Thinker algorithm provides the following advantages:

- **Flexibility:** The agent learns to plan on its own, without any handcrafted planning algorithm, allowing it to adapt to different states, environments, and models.

- **Generality:** The Thinker algorithm only dictates how an MDP is transformed, making it compatible with any RL algorithm. Specifically, one can convert any model-free RL algorithm into a model-based RL algorithm by switching to the Thinker-augmented MDP.

- **Interpretability:** We can visualize the agent's plan prior to its execution, as depicted in Fig 1. This provides greater insight compared to scenarios where the neural network internalizes the model and plans in a black box fashion [14].

- **Aligned objective:** Both the real and imaginary actions are trained using the same rewards from the environment, ensuring that the objectives of planning and acting align.

- **Improved learned model:** We introduce a novel combination of architecture and feature loss for the model, which is designed to prioritize the learning of task-relevant features and enable visualization at the same time.

## 2 Related Work

Various approaches exist for incorporating planning into RL algorithms, with the majority relying on handcrafted planning algorithms. In other words, the policy underlying the interaction with the world model, termed here as the *imaginary policy*, is typically handcrafted. For example, MuZero [5], AlphaGo [15], and AlphaZero [7] employ MCTS. VPN [16], TreeQN and ATreeC [17] explore all possible action sequences up to a predefined depth. I2A [18] generates one rollout per action, with the first action iterating over the action set and subsequent actions following the current policy.

Learning to plan is a challenging domain, and a few works exist in this area. MCTSnets [19] is a supervised learning algorithm that learns to imitate actions from expert trajectories generated from MCTS. VIN [20] and DRC [14] are model-free approaches that utilize a special neural network architecture. VIN's architecture aims to facilitate value iteration, while DRC's architecture aims to facilitate general planning. IBP [21], which is closely related to our work, allows the agent to engage with both the model and the environment. However, several critical differences distinguish IBP from Thinker. First, in IBP, the imaginary policy is trained by backpropagating through the model, a method that is primarily suitable for continuous action sets, whereas our approach focuses on environments with discrete action sets and treats the model as a black-box tool external to the agent. Second, unlike our method, IBP does not predict future values and policies, which we have identified as crucial for complex environments (see Appendix E). Third, we have designed an augmented state representation that significantly simplifies the process of using the model by providing more than just the predicted states to the agent, as is the case with IBP.

Another research direction employs gradient updates to refine rollouts [22, 23, 24] prior to taking each real action. These methods start with a preliminary imaginary policy and use this policy to gather rollouts from the model. Subsequently, these rollouts are used to calculate gradient updates for the imaginary policy by maximizing the imaginary rewards, thus incrementally refining the rollouts' quality. In the final step, a handcrafted function is used to select a real action. For instance, previous works such as [22], [23], and [24] have proposed to maximize the rollout return using REINFORCE [25], PPO [26], and backpropagation through the model, respectively. In contrast to these works, Thinker does not employ such gradient updates to refine rollouts, as the agent has to learn how to refine rollouts and select a real action on its own. Moreover, unlike these works, Thinker trains the imaginary policy to maximize the real rewards, thus preventing model exploitation when the model is learned.

Thinker is the first work showing that an RL agent *can learn to plan with a learned world model in complex environments.* Prior related works either do not satisfy (i) learning to plan, i.e., a learned imaginary policy, (ii) using a learned model or (iii) being evaluated in a complex environment. For example, MuZero [5], VPN [16], TreeQN [17], and I2A [18] do not satisfy (i). VIN [20], DRC [14] and [22, 24] do not satisfy (ii). IBP [21] and [23] do not satisfy (iii).

We describe the connection between our algorithm and Meta-RL [27, 28], as well as generalized policy iteration [29, 30], in Appendix H.

## 3 Background and Notation

We consider a Markov Decision Process (MDP) defined by a tuple $(\mathcal{S}, \mathcal{A}, P, R, \gamma, d_0)$, where $\mathcal{S}$ is a set of states, $\mathcal{A}$ is a finite set of actions, $P : \mathcal{S} \times \mathcal{A} \times \mathcal{S} \to [0, 1]$ is a transition function representing the dynamics of the environment, $R : \mathcal{S} \times \mathcal{A} \to \mathbb{R}$ is a reward function, $\gamma \in [0, 1]$ is a discount factor, and $d_0 : \mathcal{S} \to [0, 1]$ is an initial state distribution. Denoting the state, action, and reward at time $t$ by $s_t$, $a_t$, and $r_t$ respectively, $P(s, a, s') = \Pr(s_{t+1} = s'|s_t = s, a_t = a)$, $R(s, a) = \mathbb{E}[r_{t+1}|s_t = s, a_t = a]$, and $d_0(s) = \Pr(s_1 = s)$, where $P$ and $d_0$ are valid probability mass functions. An episode is a sequence of $(s_t, a_t, r_{t+1})$, starting from $t = 1$ and continuing until reaching the terminal state, a special state where the environment ends. Letting $g_t = \sum_{k=t}^{\infty} \gamma^{k-t} r_k$ denote the infinite-horizon discounted return accrued after acting at time $t$, we are interested in finding, or approximating, a *policy* $\pi : \mathcal{S} \times \mathcal{A} \to [0, 1]$, such that for any time $t \geq 1$, selecting actions according to $\pi(s, a) = \Pr(a_t = a|s_t = s)$ maximizes the expected return $\mathbb{E}[g_{t+1}|\pi]$. The value function for policy $\pi$ is $v^\pi$ where for all $s \in \mathcal{S}$, $v^\pi(s) = \mathbb{E}[g_{t+1}|s_t = s, \pi]$. We adopt the notation $(s_t, a_t, r_{t+1}, s_{t+1})$ for representing transitions, as opposed to $(s_t, a_t, r_t, s_{t+1})$, which facilitates a clearer description of the algorithm. In this work, we only consider an MDP with a discrete action set.

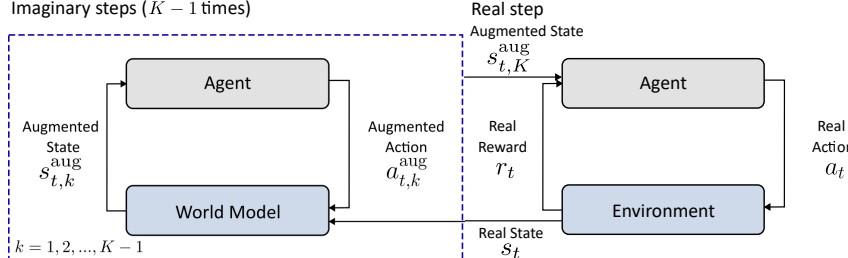

Figure 2: Illustration of a stage in the augmented MDP, which is composed of $K-1$ imaginary steps and one real step. The environment outputs a state $s_t$ that sets the root state of the world model. The agent interacts with the world model $K-1$ times with augmented actions. In each imaginary step, the agent receives an augmented state $s_{t,k}^{\text{aug}}$, which contains the latest outputs from the world model. Finally, the agent acts in the real environment for a single step based on $s_{t,K}^{\text{aug}}$.

## 4 Algorithm

The Thinker algorithm transforms a given MDP, denoted as $\mathcal{M} = (\mathcal{S}, \mathcal{A}, P, R, \gamma, d_0)$, into an augmented MDP, denoted as $\mathcal{M}^{\text{aug}} = (\mathcal{S}^{\text{aug}}, \mathcal{A}^{\text{aug}}, P^{\text{aug}}, R^{\text{aug}}, \gamma^{\text{aug}}, d_0^{\text{aug}})$. We call $\mathcal{M}$ the real MDP, and $\mathcal{M}^{\text{aug}}$ the augmented MDP. Besides the real MDP, we assume that we are given (i) a model in the form of $(\hat{P}, \hat{R})$ that is an approximate version of $(P, R)$ and is deterministic, and (ii) a *base policy* $\hat{\pi} : (\mathcal{S}, \mathcal{A}) \to [0, 1]$ and its value function $\hat{v} : \mathcal{S} \to \mathbb{R}$. The base policy can be any policy, which helps to simplify the search and evaluation of rollouts for the agent. We will show how both the model and base policy can be learned from scratch in Section 4.4, but for now we assume that both are given.

### 4.1 Augmented Transitions

To facilitate interaction with the model, we add $K-1$ steps before each step in the real MDP, where $K \geq 1$ and denotes the *stage length*. We typically use $K = 20$ in the experiments. The first $K-1$ steps are called imaginary steps, as actions in these steps are passed to the model instead of the real MDP. The step that follows the imaginary step is called the real step, as the action selected is passed to the real MDP. These $K$ steps in the augmented MDP, including $K-1$ imaginary steps and one real step, constitute a single *stage* in the augmented MDP. An illustration of a stage is shown in Fig 2. Each episode in the augmented MDP is composed of multiple stages, with a single stage corresponding to one step in the real MDP. The augmented MDP terminates if and only if the underlying real MDP terminates. We use $k \in \{1, 2, ..., K\}$ to denote the augmented step within a stage, and $t \in \{1, 2, ...\}$ to denote the current stage. We omit the subscript $t$ where it is not pertinent.

Our next step is to determine the form of the augmented transition. We can view the model search as tree traversal, with the tree's root node corresponding to the current real state $s$. We consider a simple method of traversing in a tree. At each imaginary step, the agent decides which child node to visit by imaginary action $\tilde{a} \in \mathcal{A}$ and also whether to reset the model to the root node by reset action $\tilde{a}^r \in \{0, 1\}$. In other words, the imaginary action unrolls the model by one step while the reset action sets the model back to the root node. We also impose a maximum search depth, $L$, in the algorithm. This ensures that the agent is forced to reset once its search depth exceeds $L$, as a learned model may not be accurate for a large search depth. An illustration of the tree traversal is shown in Figure 3.

After the imaginary steps, the agent chooses a real action $\tilde{a} \in \mathcal{A}$ in a real step, where this real action is passed to the real MDP. We can merge both the imaginary and real actions, while interpreting the reset action as an additional one, thereby defining the augmented action space by $\mathcal{A}^{\text{aug}} := \mathcal{A} \times \{0, 1\}$, and an augmented action by the tuple $a^{\text{aug}} := (\tilde{a}, \tilde{a}^r)$. During imaginary steps, $\tilde{a}$ becomes the imaginary action, while $\tilde{a}^r$ decides whether to reset the current node. During real steps, $\tilde{a}$ becomes the real action, while $\tilde{a}^r$ is not used. The pseudocode can be found in Algorithm 1.

### 4.2 Augmented State

We then consider the design of the augmented state that is passed to the agent. One straightforward method involves concatenating all predicted states in the tree. At each real step, the $K-1$ predicted

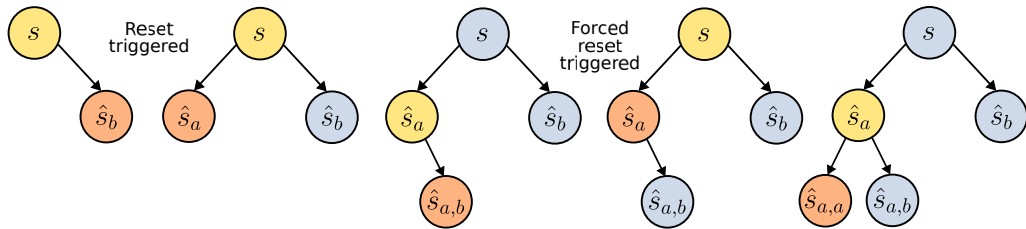

Step 1:$(b, \text{reset})$   Step 2:$(a, \text{not reset})$   Step 3:$(b, \text{not reset})$   Step 4:$(a, \text{not reset})$   Step 5:$(a, \text{reset})$

Figure 3: Illustration of tree traversal in imaginary steps, where the stage length $K = 6$ and the maximum search depth $L = 2$. Two actions, $a$ and $b$, are available. The root node corresponds to the real state $s$ underlying the stage, and the child nodes correspond to the predicted states $\hat{s}_{a_1, a_2, \dots}$ following the selection of actions $a_1, a_2, \dots$ from state $s$. The augmented actions in the $K - 1 = 5$ imaginary steps are depicted beneath the tree. The first action unrolls the model, whereas the second action determines whether the current node is reset to the root node. The yellow and the orange nodes indicate the current node before and after unrolling (before reset is applied), respectively. A forced reset is triggered on the $3^{\text{rd}}$ imaginary step, as the maximum search depth has been reached.

states serve as input, informing the selection of the real action. However, this method presents two significant challenges: (i) the high dimensionality of the predicted states makes learning to search and evaluate rollouts difficult, and (ii) the concatenation effectively multiplies the state's dimension by $K$, further complicating the learning of rollout summarization. To mitigate these issues, we propose a compact way of representing the tree in a low-dimensional vector. First, we consider how to encode a node compactly, followed by how to use these node representations to represent the tree.

To encode a node, we include the reward, the base policy and its value at the node. Moreover, we provide certain *hints* that summarize the rollouts. Taking inspiration from MCTS, we consider three hints: mean rollout return, maximum rollout return, and visit count.

For a node $i_0$ and its $n$-level descendant node $i_n$, we define the *rollout return* from node $i_0$ to node $i_n$ through the path $(i_0, i_1, i_2, ..., i_n)$ as:

$$g_{i_0, i_n} := \hat{r}_{i_0} + \gamma \hat{r}_{i_1} + \gamma^2 \hat{r}_{i_2} + ... \gamma^n \hat{r}_{i_n} + \gamma^{n+1} \hat{v}_{i_n}, \tag{1}$$

where $\hat{v}_i$ denotes the value function applied on the node $i$ and $\hat{r}_i$ denotes the predicted reward upon reaching node $i$. That is, $\hat{v}_i := \hat{v}(\hat{s}_i)$, and $\hat{r}_i := \hat{R}(\hat{s}_{\text{parent\_node}(i)}, a_{\text{parent\_edge}(i)})$.

For a given node $i$, we can compute $g_{i,j}$ for all its descendant nodes $j$ that are visited in the tree. We also define $g_{i,i} := \hat{r}_i + \gamma \hat{v}_i$. We define the mean rollout return $g_i^{\text{mean}}$ and the maximum rollout return $g_i^{\text{max}}$ as the mean of $g_{i,j}$ and the maximum of $g_{i,j}$ over all descendant nodes $j$ (including itself) that are visited respectively. The mean and maximum rollout returns of a node's children are useful in guiding actions. For example, the mean rollout return of the child $i$ of a node $k$ gives a rough estimate of the average estimated return that can be collected following action $i$ from node $k$. The mean rollout return is also equivalent to $Q(s, a)$ used in the UCB formula of MCTS. We use $g_{\text{child}(i)}^{\text{mean}}$ and $g_{\text{child}(i)}^{\text{max}}$ to denote the vector of mean and maximum rollout returns for each child of a node $i$.

The visit count of a node, which represents the number of times the agent traverses that node, may also be useful. For example, a child node that has never been visited may be worth exploring more. We use $n_{\text{child}(i)} \in \mathbb{R}^{|\mathcal{A}|}$ to denote the vector of visit count for each possible child of a node $i$.

Altogether, we represent a node $i$ by a vector $u_i \in \mathbb{R}^{5|\mathcal{A}|+2}$, called *node representation*, that contains:

1. $\hat{r}_i, \hat{v}_i \in \mathbb{R}, \hat{\pi}_i \in \mathbb{R}^{|\mathcal{A}|}$: The core statistics, composed of the reward, the value, and the base policy of the node $i$. Here, $\hat{\pi}_i := (\hat{\pi}(s_i, a_1), \hat{\pi}(s_i, a_2), ..., \hat{\pi}(s_i, a_{|\mathcal{A}|}))$.

2. $\text{one\_hot}(a_i) \in \mathbb{R}^{|\mathcal{A}|}$: The one-hot encoded action that leads to node $i$.

3. $g_{\text{child}(i)}^{\text{mean}}, g_{\text{child}(i)}^{\text{max}}, \frac{1}{K} n_{\text{child}(i)} \in \mathbb{R}^{|\mathcal{A}|}$: The hints, composed of mean rollout return, maximum rollout return and visit counts of the child nodes normalized by $K$.

We only include the root node's representation $u_{\text{root}}$ and the current node's representation $u_{\text{cur}}$ to represent the tree (we use the subscript $_{\text{root}}$ to denote the root node, which is unchanged throughout a

stage, and the subscript $_{\text{cur}}$ to denote the current node *before reset*). The former guides the selection of real actions, while the latter guides the selection of imaginary actions. Besides $u_{\text{root}}$ and $u_{\text{cur}}$, we also add auxiliary statistics, such as the current step in a stage, in the representation. Further details can be found in Appendix A. Combined, we represent the tree with a fixed-size vector $u_{\text{tree}} \in \mathbb{R}^{10|\mathcal{A}|+K+9}$, called *tree representation*. We may also include the real state $s$ or the current node's predicted state $\hat{s}_{\text{cur}}$ in the augmented state, providing a richer context for the agent.

The tree representation is designed to simplify various planning procedures. In searching, the base policy and visit count help the identification of promising, yet unvisited rollouts. In evaluation, rather than simulating until a terminal state, value estimates can be utilized to assess the potential of a given state. In summarization, mean and maximum rollout returns consolidate the outcomes for the current node. Finally, in execution, the hints at the root node provide a compact summary of all rollouts. Compared to the concatenated predicted states, which have a dimension of $K|\mathcal{S}|$, the tree representation, typically with fewer than 100 dimensions, is much easier to learn from.

The tree representation enables an agent to execute various common planning algorithms, such as $n$-step exhaustive search and MCTS, without requiring memory. Employing an agent with memory, such as an actor-critic algorithm with an RNN, may facilitate learning a broader class of planning algorithms, as the agent can access previous node representations. In principle, an agent with memory can learn to compute the hints or other ways of summarizing rollouts by itself.

## 4.3  Augmented Reward

To align the agent's goals in the augmented MDP with those in the real MDP, we adopt the following augmented reward function: no reward is given for imaginary steps, and the reward for real steps is identical to the real reward from the real MDP. In other words, $r_k^{\text{aug}} = 0$ for $k \leq K$ and $r_{K+1}^{\text{aug}} = r$, where $r$ denotes the real reward. We set the augmented discount rate $\gamma^{\text{aug}} := \gamma^{\frac{1}{K}}$ to account for the $K-1$ imaginary steps preceding one real step, ensuring that the return in the augmented MDP equals the return in the real MDP.

We also experimented with the use of an auxiliary reward, called the *planning reward*, to mitigate the sparse reward in the augmented MDP. The core idea is to encourage the agent to maximize the maximum rollout return $g_{\text{root}}^{\text{max}}$, but we later found that it only provided a minimal increase in initial learning speed, indicating that the real rewards are sufficient for learning to plan. The details of the planning reward can be found in Appendix G.

This completes the description of the augmented MDP. One may view an RL agent acting on this augmented MDP with a fixed base policy as performing one step of policy improvement in generalized policy iteration, but the policy improvement operator is learned instead of handcrafted. Experiments show that RL agents can indeed learn to improve the base policy (see Appendix H). As such, we only need to project the improved base policy back to the base policy space to complete the full cycle of generalized policy iteration, which will be discussed next.

## 4.4  Learning the World Model and the Base Policy

Until now, we assume that we are provided (i) the model $(\hat{P}, \hat{R})$ and (ii) a base policy $\hat{\pi}$ and its value $\hat{v}$. To remove both assumptions, we propose employing a single, unified model that assumes the roles of $\hat{P}, \hat{R}, \hat{\pi}$, and $\hat{v}$, learning directly by fitting the real transitions. Specifically, we introduce a novel model, the *dual network*, designed for this purpose.

The dual network consists of two RNNs which we refer to as the *state-reward network* and the *value-policy network*. The state-reward network's input consists of the root state $s_t$ and an action sequence $a_{t+1:t+L}$, and the network predicts future states $\hat{s}_{t+1:t+L}$, rewards $\hat{r}_{t+1:t+L}$, and termination probability $\hat{p}_{t+1:t+L}^d$. Meanwhile, the value-policy network takes both the state-reward network's input and its predicted states $\hat{s}_{t+1:t+L}$ as inputs, and the network predicts future policies $\hat{\pi}_{t:t+L}$ and values $\hat{v}_{t:t+L}$. We can also view the dual network as an RNN with $h = (h^{sr}, h^{vp})$ being the hidden state, where $h^{sr}$ and $h^{vp}$ denote the hidden states of the two sub-RNNs. The relevant statistics required in the augmented state can be computed by unrolling the dual network using the root node and the action sequence leading to the current node.

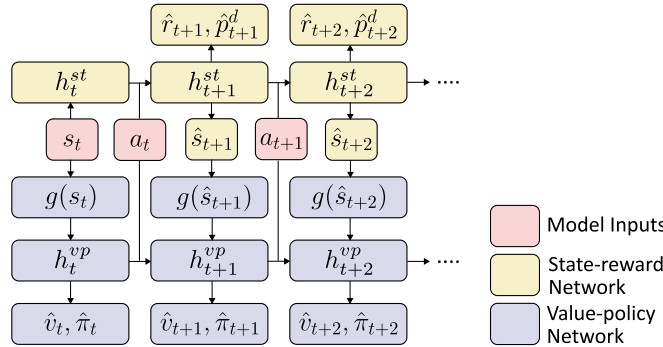

Figure 4: Illustration of the dual network's architecture, composed of the state-reward and the value-policy network. See Equations 2 and 3 for the loss functions and Appendix B for training details.

To train the model, we first store all real transitions $(s_t, a_t, r_{t+1}, d_{t+1})$ in a replay buffer, which corresponds to the state, action, reward, and termination indicator. We then sample a sequence of real transitions with length $L + 1$ from the buffer. The two sub-RNNs in the dual network are trained separately. For the state-reward network, the following loss is used:

$$\mathcal{L}^{sr} := \sum_{l=1}^{L} \left( c^r (\hat{r}_{t+l} - r_{t+l})^2 + c^d d_{t+l} \log \hat{p}_{t+l}^d + c^s \mathcal{L}^{\text{state}}(\hat{s}_{t+l}, s_{t+l}) \right), \tag{2}$$

where $c^r, c^d, c^s \geq 0$ are hyperparameters that modulate the loss strength. While a straightforward choice for the state prediction loss metric, $\mathcal{L}^{\text{state}}$, is the L2 loss – defined as $\mathcal{L}^{\text{state}}(s', s) = ||s' - s||_2^2$ – this approach could cause the model to allocate capacity towards predicting non-essential features. To address this issue, we suggest using the feature loss instead of the L2 loss. Specifically, $\mathcal{L}^{\text{state}}(s', s) = ||g(s') - g(s)||_2^2$, where $g$ represents the encoding function of the value-policy network. As this encoding function concentrates on features relevant to action, the feature loss encourages the model to prioritize predicting the most crucial features.

For the value-policy network, the following loss is used:

$$\mathcal{L}^{vp} := \sum_{l=0}^{L} \left( c^v (\hat{v}_{t+l} - \hat{v}_{t+l}^{\text{target}})^2 + c^\pi \text{one\_hot}(a_{t+l})^T \log \hat{\pi}_{t+l} \right), \tag{3}$$

where $c^v, c^\pi \geq 0$ are hyperparameters and $\hat{v}_{t+l}^{\text{target}}$ denotes the multi-step return. One may also use the real action distribution $\pi_t$ place of one\_hot$(a_{t+l})$ for a better learning signal, though this requires passing the real action distribution instead of only the actions to the augmented MDP.

The design choice of the dual network aims to improve data efficiency while also ensuring inaccurately predicted states do not negatively affect the learning of the base policy and values. If the predicted states are inaccurate, the value-policy network, being an RNN, can learn to ignore these states, relying solely on the root state and action sequence. Conversely, if the predicted states are accurate, the value-policy network can effectively learn to make predictions based on these states. Consider, for instance, the task of predicting the value after executing several actions in Sokoban. If the predicted state suggests the level is close to completion, the predicted value should naturally be higher. Although, in theory, a standalone RNN might autonomously learn this intermediary step, in practice, it would likely require much more data. We explore the relationship of this dual network with models from other model-based RL algorithms in Appendix I.

With the dual network, the augmented state can also include the hidden state of the model at either the current node, $h_{\text{cur}}$, or the root node, $h_{\text{root}}$. These hidden states, usually in much lower dimensions, might be easier to process than the real state $s$ or the current node's predicted state $\hat{s}_{\text{cur}}$. In our experiments, we use $(u_{\text{tree}}, h_{\text{cur}})$ as the augmented state, based on the intuition that the current hidden state is likely the most informative among the others.[2] This completes the specification of the Thinker algorithm, which takes an MDP as input and transforms it into another MDP.

---

[2]Strictly speaking, using $(u_{\text{tree}}, h_{\text{cur}})$ as the augmented state leads to a POMDP instead of an MDP. One can recover an MDP by omitting the auxiliary hints in $u_{\text{tree}}$ and adding $s_{\text{root}}$ to the augmented state, but both do not help performance in our experiments.

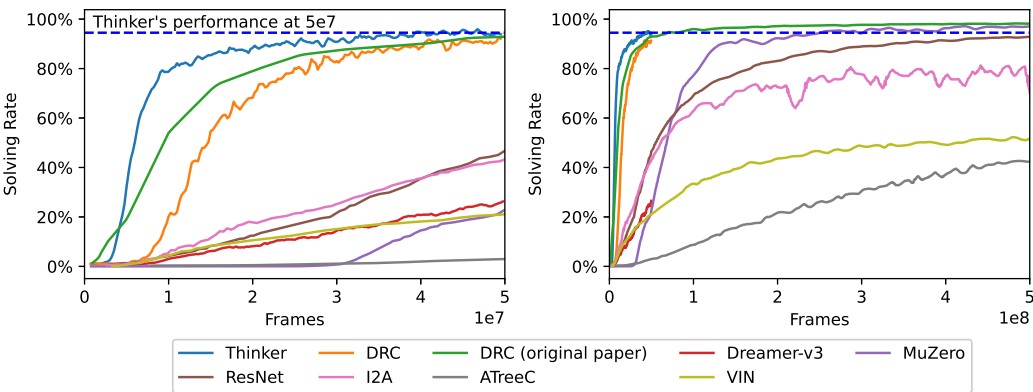

Figure 5: Running average solving rate over the last 200 episodes in Sokoban, compared with other baselines.

## 5 Experiments

For all our experiments, we train a standard actor-critic algorithm, specifically the IMPALA algorithm [31], on the Thinker-augmented MDP. Although other RL algorithms could be employed, we opted for the IMPALA due to its computational efficiency and simplicity. The actor-critic's network uses an RNN for encoding the tree representation and a convolutional neural network for encoding the model's hidden state. These encoded representations are concatenated, then passed through a linear layer to generate actions and predicted values. We set the stage length, $K$, to 20, and the maximum search depth or model unroll length, $L$, to 5. More experiment details can be found in Appendix D.

### 5.1 Sokoban

We selected the game of Sokoban [32, 18], a classic puzzle problem, as our primary testing environment due to its inherent complexity and requirement for extensive planning. In Sokoban, the objective is to push all four boxes onto the designated red target squares, as depicted in Figure 1. We used the unfiltered dataset comprising 900,000 Sokoban levels from [14]. We train our algorithm for 5e7 frames, while baselines in the literature typically train for at least 5e8 frames. Despite this discrepancy, we present learning curves plotted against frame count for direct comparison.

The learning curve for the actor-critic algorithm applied to the Thinker-augmented MDP is depicted in Fig. 5, with results averaged across five seeds. We include seven baselines: DRC [14], Dreamer-v3 [33], MuZero [5], I2A [18], ATreeC [17], VIN [20], and IMPALA with ResNet [31]. In Sokoban, DRC stands as the current state-of-the-art algorithm, outperforming others by a significant margin, likely due to its well-suited prior for the game. The 'DRC (original paper)' result is sourced directly from the paper, whereas 'DRC' denotes our replicated results.

Thinker surpasses the other baselines in terms of performance in the first 5e7 frames. At 2e7 frames, Thinker solves 88% of the levels, while DRC (original paper) solves 80% and the other baselines solve at most 21%. At 5e7 frames, Thinker solves 95 % of the levels, while DRC (original paper) solves 93% and the other baselines solve at most 45%. These results underscore the enhanced planning capabilities afforded by the Thinker-augmented MDP to an RL agent.

To understand the benefits of planning, we evaluate a policy trained with $K = 20$ on the Thinker-Augmented MDP, by testing its performance with different $K$ values in the same augmented environment during the testing phase. This variability allows us to control the degree of planning and the result is shown in Figure 6. The figure also depicts the performance of the base policy, which can be regarded as a distilled policy devoid of planning. We observe a significant performance improvement attributable to planning, even at the end of training.

The agent's behavior, visualized in Figure 1, illustrates that it learns to use the model for selecting better actions. Interestingly, it appears that the agents learn a planning algorithm that diverges from traditional $n$-step-exhaustive search and MCTS. For instance, the agent chooses real actions based on

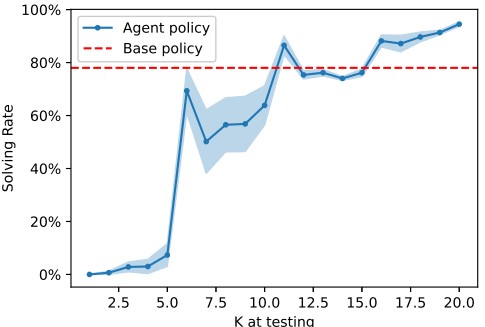

Figure 6: Final solving rate on Sokoban test levels with varying stage length $K$ during test time but trained with $K = 20$.

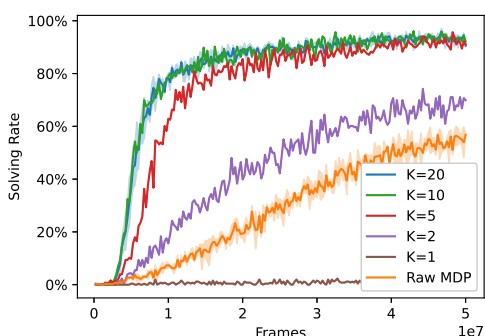

Figure 7: Running average solving rate over the last 200 episodes in Sokoban with varying stage length $K$.

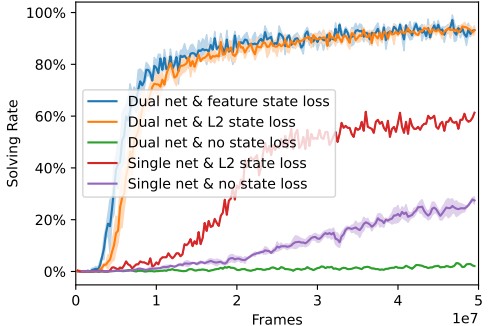

Figure 8: Running average solving rate over the last 200 episodes in Sokoban with different models.

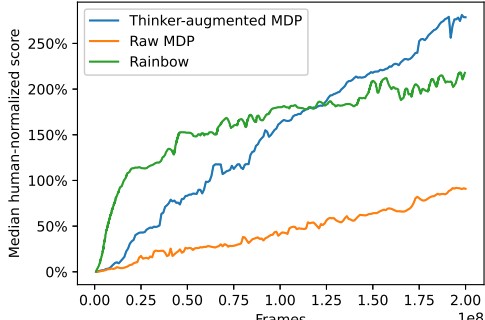

Figure 9: Median human-normalized score on Atari 2600, computed by the running average episode returns over the last 500 episodes.

both visit counts and rollout returns, and the agent learns to reset upon entering an irreversible state, contrasting the MCTS strategy of resetting at a leaf node. Further analysis on the trained agent's behavior can be found in Appendix F.

We conducted an ablation analysis to examine the impact of planning duration, varying the stage length $K$ across $\{1, 2, 5, 10, 20\}$ during both training and testing. When $K = 1$, the agent does not interact with the model at all. The outcomes of this ablation analysis are presented in Figure 7. The results suggest that $K = 10$ already gives the optimal performance. This is in stark contrast to the 2000 simulations required by MCTS to achieve good performance in Sokoban [19].

As a comparison, we train the same actor-critic algorithm on the raw MDP, employing a similar network architecture used by the model to encode the raw state. Surprisingly, the result for $K = 1$ is worse than the raw MDP. A detailed explanation of the reasons behind this phenomenon can be found in Appendix D. However, employing a marginally larger $K$, such as 2, greatly improves the performance on the Thinker-augmented MDP, surpassing that of the raw MDP.

We use the dual network as the world model, trained with feature loss in state prediction. We consider several variants of the model, including (i) a dual network that uses L2 loss in state prediction, (ii) a dual network that has no loss in state prediction, (iii) a single network that uses L2 loss in state prediction, (iv) a single network that does not predict state. The single network refers to a single RNN that predicts the relevant quantities, and (iv) is equivalent to MuZero's model. The results of these four variants are shown in Figure 8. We observe that both the dual network architecture and the state loss are critical to performance. However, the use of feature loss instead of L2 loss gives marginal performance improvement, likely because the L2 loss' prior that each pixel is equally important suits well in Sokoban. Nonetheless, this prior may not be suitable for other games like Breakout.

Further ablation studies in Appendix E offer insights into the various components of Thinker, including the role of hints in tree representation, the agent's memory, base policy, planning reward, etc. For example, either the hints *or* an RNN agent is sufficient to achieve good performance, as the hints summarize the rollouts, and an agent with memory can learn to summarize rollouts on its own. However, if both hints and memory are omitted, the agent fails to learn to plan, as it cannot recall past rollouts. In addition, if the base policy and its value are also omitted, then an RNN agent still fails to learn, suggesting the necessity of a heuristic in learning to plan.

## 5.2 Atari 2600

Finally, we test our algorithm on the Atari 2600 benchmark [34] using a 200M frames setting. The performance of the actor-critic algorithm on both the Thinker-augmented MDP and raw MDP is evaluated using the same hyperparameters as in the Sokoban experiment, but with a deeper neural network for the model and an increased discount rate from 0.97 to 0.99.

The learning curve, measured in the median human-normalized score across 57 games, is displayed in Figure 9. We evaluate the agent in the no-ops starts regime at the end of the training, and the learning curve is shown in Fig 9. The median (mean) normalized score for the actor-critic algorithm applied on the Thinker-augmented MDP and the raw MDP are 261% (1372%) and 102% (514%), respectively, underscoring the advantages of the Thinker-augmented MDP over the raw MDP. For context, the median (mean) normalized score of Rainbow [35], a robust baseline, is 223% (868%).

The advantages of the Thinker algorithm are particularly pronounced in certain types of games, especially shooting games such as `chopper command`, `seaquest`, and `phoenix`. It is plausible that the agents benefit from predicting the trajectories of fast-moving objects in these shooting games. However, in some games, particularly those where a single action does not produce significant effects, the Thinker algorithm appears to offer no advantages. For instance, in the game `qbert`, the decision to move to a different box is made only once every five or more actions. This effectively transforms any 5-step plan into a single-step plan, diminishing the algorithm's benefits. Detailed experiment results on the Atari 2600 are available in Appendix D.

Many interesting behaviors learned by the agent are visualized in `https://youtu.be/0AfZh5SR7Fk`. We also observe that the predicted frames in the video are of high quality, showing that the feature loss can lead to the emergence of interpretable and visualizable predicted frames, despite the absence of an explicit loss in the state space. We attribute this capability to using a convolutional mapping $g$ in the feature loss, as convolutions can preserve local features.

## 6 Future Work and Conclusion

While the Thinker algorithm offers many advantages, it also presents several limitations that provide potential areas for future research. Firstly, the Thinker algorithm carries a large computational cost. A single step in the original MDP corresponds to $K$ steps in the augmented MDP, in addition to the overhead of model training. Secondly, the algorithm currently enforces rigid planning, requiring the agent to roll out from the root state and restricts it to planning for a fixed number of steps. Thirdly, the algorithm employs a deterministic model to facilitate tree traversal, which may not work well for a stochastic environment.

Future work could focus on addressing these limitations, such as enabling more flexible planning steps and integrating uncertainty into the model. Additionally, exploring the application of the algorithm in multi-environment settings is a promising direction. We hypothesize that the advantages of a learned planning algorithm over a handcrafted one will become more pronounced in such scenarios. Exploring how other RL algorithms perform within the Thinker-augmented MDP is an additional direction for future work.

The history of machine learning research tells us that learned approaches often prevail over handcrafted ones. This transition becomes especially pronounced when large amounts of data and computational power are available. In the same vein, we surmise that learned planning algorithms will eventually surpass handcrafted planning algorithms in the future.

## Acknowledgement

This work was performed using resources provided by the Cambridge Service for Data Driven Discovery (CSD3) operated by the University of Cambridge Research Computing Service (www.csd3.cam.ac.uk), provided by Dell EMC and Intel using Tier-2 funding from the Engineering and Physical Sciences Research Council (capital grant EP/T022159/1), and DiRAC funding from the Science and Technology Facilities Council (www.dirac.ac.uk).

We note that author Ivan Anokhin has since moved to the Mila, Université de Montréal, after completing the primary research reflected in this paper at the University of Cambridge.

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

# A Algorithm

---

**Algorithm 1** Transition in Thinker-Augmented MDP

---

1: **Input:** real MDP: *env*, model: $\hat{P}, \hat{R}$, base policy and its value function: $\hat{\pi}, \hat{v}$.;
2: **Algorithm Parameter:** stage length $K \geq 1$, maximum search depth $L \geq 1$.
3: **Initialize** $k \leftarrow K$         ▷ Ensures that the first step is a real step
4: **procedure** STEP($\tilde{a}, \tilde{a}^r$)         ▷ $\tilde{a}$ denotes imaginary/real action, $\tilde{a}^r$ denotes reset action
5:     **if** $k = K$ **then**         ▷ Real step
6:          $k, l \leftarrow 1, 0$         ▷ Initialize step number and search depth
7:          $r, s, d \leftarrow env.step(\tilde{a})$         ▷ Compute reward, state and termination for the real MDP
8:          $s_{\text{root}} \leftarrow s$         ▷ Record the root node
9:          $s_{\text{cur}} \leftarrow s_{\text{root}}$         ▷ Set the current node to be the root node
10:          $\hat{r}_{\text{root}} \leftarrow 0$         ▷ No predicted rewards for the root node
11:     **else**         ▷ Imaginary step
12:          $k, l \leftarrow k + 1, l + 1$
13:          **if** $l \geq L$ **then**         ▷ Check if search depth exceeds maximum search depth $L$
14:              $\tilde{a}^r \leftarrow 1$         ▷ Override reset
15:          **end if**
16:          $\hat{r}_{\text{cur}} \leftarrow \hat{R}(s_{\text{cur}}, \tilde{a})$         ▷ Compute the predicted reward
17:          $s_{\text{cur}} \leftarrow \hat{P}(s_{\text{cur}}, \tilde{a})$         ▷ Compute the new current node by unrolling the model
18:          $r \leftarrow 0$         ▷ No rewards for imaginary steps
19:          $d \leftarrow False$         ▷ No termination for imaginary steps
20:     **end if**
21:     $\hat{\pi}_{\text{cur}}, \hat{v}_{\text{cur}} \leftarrow \hat{\pi}(s_{\text{cur}}), \hat{v}(s_{\text{cur}})$         ▷ Compute the base policy and its value
22:     Compute $s^{\text{aug}}$ using $s_i, \hat{r}_i, \hat{\pi}_i, \hat{v}_i$ where $i \in \{\text{root}, \text{cur}\}$ as described in Section A.1
23:     **if** $\tilde{a}^r = 1$ **then**         ▷ Reset to the root node if reset is triggered
24:          $l \leftarrow 0$
25:          $s_{\text{cur}} \leftarrow s_{\text{root}}$
26:     **end if**
27:     **return** $r, s^{\text{aug}}, d$         ▷ Return rewards, state, and termination for the augmented MDP
28: **end procedure**

---

Algorithm 1 implements the step function of the Thinker-augmented MDP, assuming we have access to a given model $(\hat{P}, \hat{R})$, a base policy $\hat{\pi}$, and its value $\hat{v}$. We assume $\hat{P}$ is deterministic and use $s' = \hat{P}(s, a)$ to denote the predicted next state $s'$ from state $s$ and action $a$.

To compute the hints (i.e. mean rollout return, max rollout return, and visit counts) in the tree representation, we need to maintain a tree with each node storing the predicted reward $\hat{r}_i$, value $\hat{v}_i$, and visit count $n_i$, but the details of updating the tree are omitted in the pseudocode for clarity.

It is important to note that since $\hat{P}$ is deterministic, one can store the computed node in a buffer to avoid re-computing it when the agent traverses the same node again within the same stage.

In the full Thinker-augmented MDP, both the model and the base policy with its values are learned by an RNN model. The code only needs slight modifications as follows:

1. After line 7, we insert the real transition in the form of $(s_{\text{old}}, \tilde{a}, r, d)$, where $s_{\text{old}}$ denotes the previous real state, into a replay buffer. A parallel thread trains the model by sampling from the replay buffer.

2. A node is represented by the hidden state of the RNN model, not the real state. Consequently, line 8 should initialize the root node with the first hidden state $h$ of the RNN model. Line 17 updates the new current node to the new hidden state by unrolling the RNN model with the current node and the latest action. Lines 16 and 21 compute the reward, policy, and value as predicted by the RNN model.

**Implementation details** Several key implementation details are worth noting. First, when transitioning into a new stage, we carry forward the tree from the previous stage and prune the non-descendant nodes of the new root node. This approach ensures the continuity of the previous plan into the

subsequent stages. However, our experiments show that carrying the tree has minimal impact on performance. Second, for clarity, the planning reward is not shown in the pseudocode, but details can be found in Appendix G. Lastly, if the model predicts episode termination at node $i$, defined by $\hat{p}_i^d > 0.5$, we manually set the content of all its descendant nodes $j$ to $\hat{r}_j = 0$, $\hat{\pi}_j = \hat{\pi}_i$, $\hat{v}_j = 0$, $\hat{s}_j = \mathbf{0}$, and $h_j = h_i$, which represents the terminal state that is absorbing and has zero rewards and values.

## A.1 Augmented State

As discussed in Section , both the current node representation $u_{\text{cur}}$ and the root node representation $u_{\text{root}}$ are included in the tree representation. Denoting the current search depth by $d \in \{0, 1, ..., L\}$, the full tree representation $u_{\text{tree}} \in \mathbb{R}^{10|\mathcal{A}|+K+9}$ contains:

1. $u_{\text{root}}, u_{\text{cur}} \in \mathbb{R}^{5|\mathcal{A}|+2}$: The root node representation and the current node representation.
2. $g_{\text{root,cur}} \in \mathbb{R}$: Current rollout return.
3. $g_{\text{root,cur}} - \gamma^{d+1}\hat{v}_{\text{cur}} \in \mathbb{R}$: Sum of discounted rewards collected in the current rollout.
4. $g_{\text{root}}^{\max} \in \mathbb{R}$: Maximum rollout return of the root node.
5. $\gamma^{d+1} \in \mathbb{R}$: Trailing discount factor.
6. one_hot$(k) \in [0, 1]^K$: Current step in a stage, represented in one-hot encoding.
7. $i^{\text{reset}} \in [0, 1]$: Indicator of whether the current node has just been reset to the root node.

And the following five inputs can be included in the augmented state $s_{t,k}^{\text{aug}}$:

1. $u_{\text{tree}} \in \mathbb{R}^{10|\mathcal{A}|+K+9}$: The tree representation.
2. $s \in \mathcal{S}$: The real state underlying the stage.
3. $\hat{s}_{\text{cur}} \in \mathcal{S}$: The predicted real state of the current node.
4. $h_{\text{root}} \in \mathcal{H}$: The hidden state of the RNN model at the root node.
5. $h_{\text{cur}} \in \mathcal{H}$: The hidden state of the RNN model at the current node.

We used $s_{t,k}^{\text{aug}} = (u_{\text{tree}}, h_{\text{cur}})$ in our main experiments. Our code enables users to decide which inputs to include as part of the augmented state in the Thinker-augmented MDP.

## A.2 Code

The complete code is accessible at `https://github.com/stephen-chung-mh/thinker`. We implemented the Thinker-augmented MDP in batches using Cython [36], a Python language extension that optimizes performance by interfacing with C. This ensures that the primary computational burden is attributed to model training and the RL algorithm, rather than the logic flow within the augmented MDP. The code supports multi-thread processing, allowing multiple threads of Thinker-augmented MDP to share the same model.

Our implementation of the Thinker-augmented MDP aligns with the OpenAI Gym interface [10], facilitating easy integration with existing RL frameworks. The code allows customization of various parameters such as stage depth $K$, maximum search depth $L$, and the model's learning rate, among others. Included in the repository is the code for the IMPALA algorithm applied to the Thinker-augmented MDP, reproducing the experimental results presented in the paper.

# B  Model Training Procedure

## B.1  Collection of Training Samples

A transition tuple $x_t := (s_t, a_t, r_{t+1}, d_{t+1})$ is stored in a replay buffer during each real step, which corresponds to the state, action, rewards, and termination indicator. The replay buffer is a first-in-first-out buffer with a capacity of 200,000 transitions. To train the model, a batch of sequences $x_{i:i+2L}$ is randomly sampled from the buffer, where $L$ represents the model rollout length. We use a batch size of 128 and a model unroll length $L = 5$.

Consistent with MuZero [5], prioritized sampling [37] is used for sampling the sequence $x_{i:i+2L}$. Each transition tuple is assigned a priority $p_i$. Newly added transition tuples are given the maximum priority value in the buffer. During training, the priority $p_i$ of the first transition in a sampled sequence $x_{i:i+2L}$ is updated to the L1 loss of the value, $|\hat{v}_i - v_i^{\text{target}}|$. Transitions are sampled with a probability of $P(i) = \frac{p_i^\alpha}{\sum_j p_j^\alpha}$, where $\alpha = 0.6$. Each sampled transition sequence's loss is weighted by the importance weight $w_i = (\frac{1}{NP(i)})^\beta$, where $N$ is the total number of transition tuples in the buffer, and $\beta$ is annealed from $0.4$ to $1.0$ throughout training.

We employ a replay ratio of 6, implying that model training is paused if the ratio of the total number of transitions trained on the model to the total collected transitions exceeds 6.

## B.2  Additional Details for Training the Model

The state-reward network and the value-policy network are trained separately, which leads to the following implications. First, when training the state-reward network, the $g$ within feature loss for state prediction is not optimized since it belongs to the value-policy network. This prevents $g$ from degeneration. Second, when training the value-policy network, gradients cannot flow through $\hat{s}_{t+l}$ towards the state-reward network since $\hat{s}_{t+l}$ is considered as the input to the value-policy network. This ensures that the predicted state aligns with the actual future state.

The multi-step return $\hat{v}_{t+l}^{\text{target}}$ in Equation 3 is defined by:

$$\hat{v}_{t+l}^{\text{target}} := r_{t+l+1} + \gamma r_{t+l+2} + ... + \gamma^{2L-l} r_{t+2L} + \gamma^{2L-l} g_{\text{root},t+2L}^{\text{mean}}. \tag{4}$$

Instead of using the predicted value $\hat{v}_{t+2L}$, we opted for the mean rollout return (at the end of the stage) $g_{\text{root},t+2L}^{\text{mean}}$ as the bootstrap target[3]. We found that it yields more stable training in our experiments. This stability likely stems from averaging across multiple rollouts starting from $s_{t+2L}$, rather than relying directly on the predicted value.

---

[3] $g_{\text{root}}^{\text{mean}}$ does not contain the reward upon arriving the root node, as $\hat{r}_{\text{root}}$ is manually set to 0 in Algorithm 1.

## C Network Architecture

In this section, we describe the neural networks that we use for the model and the actor-critic network.

### C.1 Model

For Sokoban, the shape of the real state, $s_t$, is (3, 80, 80), while for Atari 2600, the real state's shape is (12, 80, 80) due to the stacking of four frames together. Before inputting the real state $s_t$ to the encoder, we concatenate it with the one-hot encoding of the action $a_{t-1}$, tiled to (80, 80), channel-wise. The encoder architecture is the same for both the state-reward and value-policy networks, structured as follows:

- Convolution with 64 output channels and stride 2, followed by a ReLu activation.
- $D$ residual blocks, each with 64 output channels.
- Convolution with 128 output channels and stride 2, followed by a ReLu activation.
- $D$ residual blocks, each with 128 output channels.
- Average pooling operation with stride 2.
- $D$ residual blocks, each with 128 output channels.
- Average pooling operation with stride 2.

All convolutions use a kernel size of 3. The resulting output shape for both Sokoban and Atari 2600 is (128, 6, 6). We use $D = 1$ for Sokoban and $D = 2$ for Atari 2600.

We denote $g^{sr}(s_t)$ and $g(s_t)$ as the encoder's output of $s_t$ for the state-reward network and the value-policy network respectively, and the dependency on $a_{t-1}$ is omitted for clarity.

For the state-reward network, we define the hidden state $h_t^{sr} := g^{sr}(s_t)$. To obtain $h_{t+l+1}^{sr}$ from $(h_{t+l}^{sr}, a_{t+l})$ with $l \geq 0$, we first concatenate $h_{t+l}^{sr}$ with the one-hot encoding of the action $a_{t+l}$, tiled to (6, 6), channel-wise. This concatenated output is passed to the unrolling function, which consists of $4D + 1$ residual blocks, each having an output channel of 128. The output of the unroll function is $h_{t+l+1}^{sr}$.

For the value-policy network, in addition to the previous hidden state, the unrolling function should also take the encoded predicted frame $g(\hat{s}_{t+l})$ as input, so the unrolling procedure is slightly modified. First, we define $h_{t-1}^{vp}$ as a zero vector of shape (128, 6, 6). To obtain $h_{t+l}^{vp}$ from $(h_{t+l-1}^{vp}, a_{t+l-1}, g(\hat{s}_{t+l}))$ with $l \geq 0$ (replace $\hat{s}_{t+l}$ with $s_{t+l}$ when $l = 0$ here), we first concatenate all three together channel-wise, with $a_{t+l-1}$ tiled to (6, 6). This concatenated output is passed to the unrolling function, which consists of $4D + 1$ residual blocks, each having an output channel of 128. The output of the unroll function is $h_{t+l}^{vp}$.

To compute the predicted quantities, we apply a prediction function to $h_{t+l}$ for both state-reward and value-policy networks. The predicted quantities for the state-reward network are rewards $\hat{r}_{t+l}$ and termination probability $\hat{p}_{t+l}^d$, while the predicted quantities for the value-policy network are values $\hat{v}_l$ and policies $\hat{\pi}_l$. The architecture of this prediction function is the same for both networks and is structured as follows:

- Convolution with 64 output channels and stride 1, followed by a ReLu activation.
- Convolution with 32 output channels and stride 1, followed by a ReLu activation.
- Flatten the 3D vector to a 1D vector.
- Separate linear layers for each predicted quantity.

In the case of probability outputs, namely the predicted termination probability $\hat{p}_{t+l}^d$ and the predicted policy $\hat{\pi}_{t+l}$, we pass the output of the linear layer through a softmax layer to ensure a valid probability distribution. In addition, the weights and biases of the final linear layers are initialized to zero to stabilize initial training.

For the state-reward network, we compute the predicted frame, $\hat{s}_{t+l}$ for $l \geq 1$, by applying a decoder on $h_{t+l}^{sr}$ for $l \geq 1$. The decoder's structure is as follows:

- $D$ residual blocks, each with 128 output channels.
- ReLu activation followed by a transpose convolution with 128 output channels and stride 2.
- $D$ residual blocks, each with 128 output channels.
- ReLu activation followed by a transpose convolution with 128 output channels and stride 2.
- ReLu activation followed by a transpose convolution with 64 output channels and stride 2.
- $D$ residual blocks, each with 64 output channels.
- ReLu activation followed by a transpose convolution with 3 output channels and stride 2.

All convolutions use a kernel size of 4. The decoder outputs a tensor of shape $(3, 84, 84)$. In Atari, we concatenate this output with the three most recent true or predicted frames in $s_{t+l-1}$ for $l = 1$ or $\hat{s}_{t+l-1}$ for $l > 1$, resulting in a final decoded predicted state with shape $(12, 84, 84)$. This approach is adopted to prevent the model from redundantly replicating previous inputs, as the prior three frames are already present in the input or have been predicted in preceding steps.

## C.2 Actor-critic

The augmented state consists of the model's hidden states $h = [h_k^{sr}, h_k^{vp}]$, a 3D vector of shape $(256, 6, 6)$, and the tree representation $u_{\text{tree}}$, a flat vector of shape $(10|\mathcal{A}| + K + 9)$. The model's hidden state is processed by a convolutional neural network (CNN), structured as:

- Convolution with 64 output channels and stride 1, followed by a ReLu activation.
- 2 residual blocks, each with 64 output channels.
- Convolution with 64 output channels and stride 1, followed by a ReLu activation.
- 2 residual blocks, each with 64 output channels.
- Convolution with 32 output channels and stride 1, followed by a ReLu activation.
- 2 residual blocks, each with 32 output channels.
- Flatten the 3D vector to a 1D vector.
- A linear layer with output size 256, followed by a ReLu activation.

The tree representation is processed by a special recurrent neural network (RNN) architecture, which combines the Long Short-Term Memory (LSTM) [38] with the attention module of the Transformer [39]. This RNN architecture is necessitated by the requirements of the Thinker-augmented MDP, which relies on long-term memory capabilities for plan execution. For instance, a 5-step plan in the augmented MDP requires the agent to maintain the plan in memory for $5K = 100$ steps, a demand that exceeds the capabilities of a traditional LSTM. However, using Transformers solely could lead to instability in RL [40] due to their disregard of the Markov property in MDP as they process inputs across all steps in the same manner. Therefore, our approach merges the LSTM with the attention module of the Transformer to allow for extended memory access while preserving the stability of the training process.

In detail, the RNN processes the tree representation as follows:

- A linear layer with output size 128, followed by a ReLu activation.
- A LSTM-Attention network (explained in detail below) with output size 128.
- A linear layer with output size 128, followed by a ReLu activation.

Finally, the input for the final layer is obtained by concatenating the following:

1. The output from the CNN processing the model's hidden state.
2. The output from the RNN processing the tree representation.
3. The one-hot encoding of the last actions (real action, imaginary action, and reset action).
4. The last rewards (both the real reward and the planning reward).

This input is passed to separate linear layers to generate the final output of the actor-critic network, which includes predicted values and logits of the policies for real actions, imaginary actions, and reset actions. We use a separate head for computing the policies for real and imaginary actions, and select the respective logits based on whether the current step is a real step or an imaginary step. To generate valid probabilities, the logits predicted by the linear layers are fed into a softmax layer. Additionally, an L2 regularization cost of 1e-9 is added for the final layer's input and 1e-6 for the real action's logits.

**LSTM-Attention Cell**   This section details the LSTM-Attention cell, which serves as the building block of the RNN. The notation here is distinct and should not be conflated with the notation in other sections. The LSTM-Attention cell, while akin to a conventional LSTM, incorporates an additional input from the attention module, multiplied by a gate variable. The LSTM-Attention cell is denoted by:

$$(h_t, c_t, K_t, V_t) = \text{LSTM\_Attn}(x_t, h_{t-1}, c_{t-1}, K_{t-1}, V_{t-1}), \tag{5}$$

where $x_t$ denotes the input to the cell, $h_t$ denotes the hidden state, $c_t$ denotes the input activation, $K_t$ and $V_t$ denotes the key and value vector for the attention module, respectively. The function LSTM_Attn is defined as:

$$f_t = \sigma(W_f x_t + U_f h_{t-1} + b_f), \tag{6}$$
$$i_t^c = \sigma(W_{ic} x_t + U_{ic} h_{t-1} + b_{ic}), \tag{7}$$
$$i_t^a = \sigma(W_{ia} x_t + U_{ia} h_{t-1} + b_{ia}), \tag{8}$$
$$o_t = \sigma(W_o x_t + U_o h_{t-1} + b_o), \tag{9}$$
$$a_t, K_t, V_t = \text{attn}(x_t, , K_{t-1}, V_{t-1}), \tag{10}$$
$$\tilde{c}_t = \tanh(W_c x_t + U_c h_{t-1} + b_c), \tag{11}$$
$$c_t = f_t \odot c_{t-1} + i_t^c \odot \tilde{c}_t + i_t^a \odot \tanh(a_t) \tag{12}$$
$$h_t = o_t \odot \tanh(c_t), \tag{13}$$

where $\{W_f, W_{ic}, W_{ia}, W_o, W_c, U_f, U_{ic}, U_{ia}, U_o, U_c, b_f, b_{ic}, b_{ia}, b_o, b_c\}$ are parameters with output shape $d_m = 128$ and $\odot$ denotes element-wise product. The only differences compared to a standard LSTM cell are Equations 8, 10, and 12. Equation 8 computes the input gate for the attention module outputs, Equation 10 represents the attention module which will be described below, and Equation 12 adds the gated output from the attention module to the input activation.

The attention function $\text{attn}(x_t, K_{t-1}, V_{t-1})$ is based on the multi-head attention module used in Transformer, with the following modifications: (i) Attention is restricted to the last $d_t = 40$ steps, equivalent to two stages. This is because maintaining an excessively long memory might complicate learning due to the Markov assumption of MDP. (ii) A learned relative positional encoding, akin to Transformer XL [41], is used to facilitate attention to steps relative to the current step.

For clarity, we first describe a single-head attention module. We denote $q_t \in \mathbb{R}^{d_e}$, $k_t \in \mathbb{R}^{d_e}$, $v_t \in \mathbb{R}^{d_e}$ as the query, key, and value at step $t$, respectively, where $d_e = 16$ is the dimension of the embedding. The input $K_{t-1} \in \mathbb{R}^{d_t, d_e}$ and $V_{t-1} \in \mathbb{R}^{d_t, d_e}$ represent the preceding $d_t$ keys and values, defined as follows:

$$K_{t-1} := [k_{t-d_t}, k_{t-d_t+1}..., k_{t-1}]^T, \tag{14}$$
$$V_{t-1} := [v_{t-d_t}, v_{t-d_t+1}..., v_{t-1}]^T. \tag{15}$$

We calculate the current $q_t, k_t, v_t$ via a linear projection on input $x_t$:

$$q_t = W_q x_t, \tag{16}$$
$$k_t = W_k x_t, \tag{17}$$
$$v_t = W_v x_t, \tag{18}$$

where $\{W_q, W_k, W_v\}$ are parameters with output shape $d_e$. This enables us to compute $K_t$ and $V_t$ by stacking the corresponding $k$ and $v$. Subsequently, we compute the attention vector $p_t \in \mathbb{R}^{d_t}$ as:

$$p_t = \text{softmax}\left(\frac{(K_t + U_r)q_t}{\sqrt{d_e}} + b_r\right), \tag{19}$$

where $U_r \in \mathbb{R}^{d_t, d_e}$ and $b_r \in \mathbb{R}^{d_t}$ are learnable parameters representing the relative position encoding. This differs subtly from the relative position encoding used in Transformer XL, as the fixed attention

window in our case allows us to utilize fixed-size parameters for encoding relative positions. We initialize $U_r$ and $b$ to be zero vectors, with the last entry of $b$ set to 5, reflecting the prior that the current step should be attended to more. In addition, the softmax here should be masked for steps in the previous episode.

The final output of a single-head attention module, denoted as $\tilde{a}_t \in \mathbb{R}^{d_e}$, is computed as follows:

$$\tilde{a}_t = V_t^T p_t. \tag{20}$$

For the multi-head attention module, we repeat the above single-head attention module $d_h = 8$ times and stack the outputs to yield $\tilde{a}_t^{\text{stack}} = \text{stack}([\tilde{a}_{t,1}, \tilde{a}_{t,2}, ..., \tilde{a}_{t,d_h}])$, where $\tilde{a}_t^{\text{stack}} \in \mathbb{R}^{d_m}$ and $\tilde{a}_{t,i}$ denotes the output of the $i^{\text{th}}$ single-head attention module. The final output of the attention module $a_t$ is given by applying a linear layer and layer normalization [42] on this stacked output:

$$a_t = \text{LayerNorm}(W_a \tilde{a}_t^{\text{stack}} + b_a + x_t), \tag{21}$$

where $x_t$ represents the skip connection, and $\{W_a, b_a\}$ are parameters with output size $d_m$. This concludes the description of the LSTM-Attention cell.

**LSTM-Attention Network** An LSTM-Attention Network is constructed by stacking $d_n = 3$ LSTM-Attention Cells together. We denote the network's input as $x_t$, and use the superscript $1 \leq n \leq d_n$ to represent the $n^{\text{th}}$ cell in the network. We denote the LSTM-Attention network as:

$$(h_t^{1:d_n}, c_t^{1:d_n}, K_t^{1:d_n}, V_t^{1:d_n}) = \text{LSTM\_Attn\_Net}(x_{t-1}, h_{t-1}^{1:d_n}, c_{t-1}^{1:d_n}, K_{t-1}^{1:d_n}, V_{t-1}^{1:d_n}), \tag{22}$$

defined by the iterative equation for $1 \leq n \leq d_n$:

$$(h_t^n, c_t^n, K_t^n, V_t^n) = \text{LSTM\_Attn}(\text{stack}([x_{t-1}, h_t^{n-1}]), h_{t-1}^n, c_{t-1}^n, K_{t-1}^n, V_{t-1}^n), \tag{23}$$

and we define $h_t^0 := h_{t-1}^n$, the top LSTM-Attention cell's output from the previous step. The final output of the network is the top LSTM-Attention cell's output, $h_t^n$. The input to each LSTM-Attention cell comprises both the network's input and the output from the previous LSTM-Attention cell. This design choice was inspired by DRC [14], which suggests that this recurrence method may facilitate better planning.

The rationale for integrating LSTM with the attention module in this manner stems from the following reasoning. The LSTM's hidden states can be interpreted as a form of short-term memory, primarily focusing on the current step. In contrast, the attention module acts as a long-term memory unit, as it maintains an equal computational path from all past steps. By fusing these two forms of memory, the LSTM is granted the capability to control access to long-term memory, including the ability to disregard it entirely by setting the gate variable to zero. This flexibility enables a focus on the most recent few steps while still preserving access to long-term memory, thus combining the advantages of both types of RNN. Using gradient clipping when applying the LSTM-Attention network is also recommended to stabilize training further.

A significant drawback of the proposed LSTM-Attention network is the sequential computation it requires. Unlike the Transformer, which can compute attention across all steps simultaneously, the proposed network has to compute each step one at a time due to the recurrent computation of LSTM. Consequently, this leads to a much higher computational cost. Nonetheless, this sequential computation limitation may not be as significant in an RL setting. An RL task inherently requires the agent to generate action sequentially, and this sequential computation cannot be avoided even for Transformer. Future studies could explore the potential of the proposed LSTM-Attention network in contexts beyond the Thinker-augmented MDP, such as RL tasks that require long-term memory.

# D   Experiment Details

## D.1   Hyperparameters

The hyperparameters used in all our experiments are shown in Table 4. We tune our hyperparameters exclusively on Sokoban based on the final solving rate. The specific hyperparameters we tune are: learning rates, model batch size, model loss scaling, maximum search depth or model unroll length, planning reward scaling, actor-critic clip global gradient norm, and actor-critic unroll length. In terms of preprocessing for Atari, we adhere to the same procedure as outlined in the IMPALA paper [31], with the exception that we do not apply grayscaling.

For the actor-critic baseline implemented on the raw MDP, we maintain the same hyperparameters, with two exceptions: (i) we adjust the actor-critic unroll length to 20 to ensure the same number of real transitions in an unrolled trajectory; (ii) we reduce the learning rate by half to 3e-4, after observing that the original learning rate led to unstable learning. We found that this set of hyperparameters provides optimal performance for the baseline in Sokoban, based on the final solving rate. Regarding the network architecture, we employ an encoder that mirrors the model's encoder architecture in Thinker, followed by a CNN that mirrors the actor-critic's CNN in Thinker.

## D.2   Detailed Results on Sokoban

The performance of Thinker and the baselines on Sokoban can be found in Table 1, corresponding to the learning curves shown in Figure 5. We use our own implementation for DRC [14] and open-source implementation for Dreamer-v3 [33], with the result being averaged over three seeds. MuZero's result is taken from [43]. Results of DRC (original paper), I2A, ATreeC, VIN and IMPALA with Resnet are taken from [14].

As Thinker augments the MDP by adding $K - 1$ steps for every step in the real MDP, there is an increased computational cost due to the extra K-1 imaginary steps, in addition to the added computational cost of training the model. On our workstation equipped with two A100s, the training durations for the raw MDP, DRC, and Thinker on Sokoban are around 1, 2, and 7 days, respectively.

However, when benchmarked against other model-based RL baselines, Thinker's training time is similar or even shorter. For instance, Dreamer-v3 (the open-sourced implementation) requires around 8 days for a single Sokoban run on our hardware, primarily due to its high replay ratio. MuZero, when conducting a standard number of simulations (e.g., 100), requires 10 training days. Notably, despite the Thinker's additional training overhead for RL agents, the algorithm compensates by requiring significantly fewer simulations. Furthermore, if one replaces the RNN in the actor-critic network with an MLP (as detailed in Appendix E), without compromising the performance, the computational time for a single Thinker run can be reduced to 3 days.

In Figure 7, we observed that the result for $K = 1$, which does not use any imaginary transitions, underperforms compared to the result on the raw MDP. Initially, we hypothesized that this was because the model's hidden state, rather than the real state, was included in the augmented state. Although this hidden state might be more effective for predicting future states and rewards, it potentially complicates the learning of an optimal policy. However, subsequent experiments revealed that substituting the real state for the model's hidden state did not impact performance. This outcome suggests that the additional information given by the tree representation is, in fact, *detrimental* to performance. We speculate that this counterintuitive effect is due to the tree representation providing information, such as value estimation, that are so insightful that the critic in the actor-critic algorithm overly relies on this estimation, thereby neglecting the real state and impeding the learning of a useful state representation for the actor. To investigate this speculation, we experimented with masking the tree representation at a random rate (with a probability $p = 0.20$), which restored the performance to the level of the raw MDP. This propensity of the agent to depend excessively on a single type of input shown in these experiments points to a potential avenue for subsequent research.

## D.3   Detailed Results on Atari 2600

The performance of Thinker and the baselines on Atari 2600 can be found in Table 2, corresponding to the learning curves shown in Figure 9. We include two additional baselines, UNREAL [44] and LASER [45], for comparison. The results of the baselines are taken from their original papers.

Table 3: Atari scores after 200M frames of training for the five selected games with different seeds. Up to 30 no-ops at the beginning of each episode.

|  | Seed 1 | Seed 2 | Seed 3 | Mean |
|---|---|---|---|---|
| Battle Zone | 78740.00 | 31110.00 | 34460.00 | 48103.33 |
| Double Dunk | 12.20 | 14.54 | 21.54 | 16.09 |
| Name This Game | 34079.60 | 33400.20 | 31775.80 | 33085.20 |
| Phoenix | 543987.70 | 535685.10 | 578485.30 | 552719.37 |
| Qbert | 30734.50 | 36810.25 | 37003.25 | 34849.33 |

We note that Thinker's performance on Atari 2600 lags behind that of state-of-the-art algorithms. There are several possible reasons: (i) we did not tune hyper-parameters on Atari 2600; (ii) learning to plan may require a much larger number of frames to show significance, similar to how MuZero requires 20 billion frames. As the goal of the experiments on Atari 2600 is to demonstrate the general advantage of the Thinker-augmented MDP over the raw MDP, we leave the problem of mastering Atari 2600 using Thinker for future study. In principle, more advanced RL algorithms, including the baselines, could be applied to the Thinker-augmented MDP.

The final scores and the corresponding learning curves for each individual Atari game are presented in Table 5 and Figure 11, respectively. The Rainbow results are taken from the original paper [35], where the reported score is obtained from the best agent snapshots throughout training. The learning curves of Rainbow in Figure 11 are also taken from the original paper.

To validate the reproducibility of our results, we carried out multiple runs of five selected games with three different seeds. These games include `battle zone`, `double dank`, `name this game`, `phoenix`, and `qbert`. The selection of these games was influenced by the recommendations of [46], which suggests that these five games are pivotal in predicting the median human-normalized score. The final scores for these five games are displayed in Table 3, and their individual learning curves can be found in Figure 10. The results for these five games, as presented in other figures and tables in this paper, are derived from the average of these three seed values.

## D.4   Discussion on Atari 2600's results

As stated in the main paper, Thinker demonstrates superior performance in shooting games like `chopper command`, `seaquest`, `phoenix`, `beam rider`, and `space invader`. A more general observation is that Thinker excels in environments with fast-moving objects, such as `breakout`. As depicted in `https://youtu.be/0AfZh5SR7Fk`, Thinker's model is capable of predicting the trajectories of these swift objects with near-perfect precision. We hypothesize that such predictions, essential for planning future actions, pose a challenge for an agent in the raw MDP to learn. For example, in `Breakout`, the video illustrates the agent exploring different methods of deflecting the ball at various angles, sometimes missing the ball in its imagination but performing adeptly in the

Table 1: Solving rate on Sokoban.

|  | At 2e7 frames | At 5e7 frames |
|---|---|---|
| Thinker | 88% | 95% |
| DRC [14] | 68% | 92% |
| DRC (original paper) [14] | 80% | 93% |
| Dreamer-v3 [33] | 12% | 32% |
| MuZero [5] | 0% | 21% |
| IMPALA with ResNet [31] | 14% | 45% |
| I2A [18] | 21% | 43% |
| ATreeC [17] | 1% | 3% |
| VIN [20] | 12% | 21% |

Table 2: Final human-normalized score on Atari 2600.

|  | Median | Mean |
|---|---|---|
| Thinker (Agent policy) | 261% | 1372% |
| Thinker (Base policy) | 160% | 807% |
| Raw MDP | 102% | 514% |
| Rainbow [35] | 223% | 868% |
| UNREAL [44] | 250% | 880% |
| LASER [45] | 431% | n.a. |

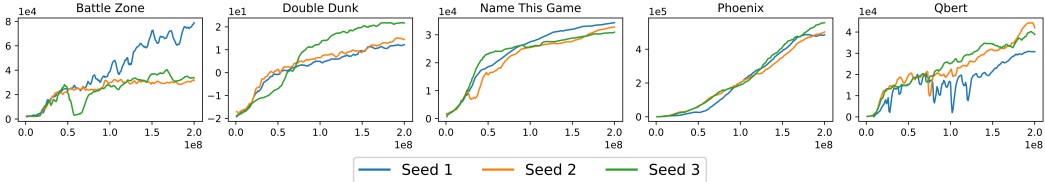

Figure 10: Learning curves for Thinker in the five selected games with different seeds. Each curve is computed as the running average of episode returns over the last 500 episodes.

actual game. Upon identifying a sequence of actions that might result in missing the ball, the agent can opt to avoid it. This adaptive decision-making and the significant performance gap between the agent policy and the base policy in these games underscore the benefits of planning in games with fast-moving objects.

It is important to highlight that applying a direct L2 loss on future states instead of a feature loss can undermine these advantages. For example, we observed that in `Breakout`, a model trained with direct L2 loss is able to predict frames of high quality, except that the ball, arguably the most important object in the game, is missing in the predicted frame. Given that these fast-moving objects typically inhabit only a few pixels and are challenging to predict, the model is prone to neglect these elements when trained using a direct L2 loss. In contrast, by using feature loss, these important objects, such as the ball in `Breakout`, are usually learned early in training.

We also observe that the agent's performance falls short in more demanding exploration games, such as `montezuma revenge` and `venture`. This is because the Thinker-augmented MDP lacks any mechanism to encourage exploration in the real environment, which results in challenges when dealing with these complex exploration games, much like an actor-critic applied directly to a raw MDP. Future research could explore methods to utilize Thinker to foster exploration, such as using model prediction loss as a curiosity reward. As exploration is a well-studied area in RL [47], many existing exploration methods could also be directly applied to the Thinker-augmented MDP.

Thinker also performs poorly in games where each action carries little impact, such as in `qbert`. Games in which the agent moves slowly, such as `bank heist` and `ms pacman`, also share this characteristic. These games are in stark contrast to games like Sokoban, where a single action can move the agent to a different grid. These games highlight a fundamental limitation of the Thinker-augmented MDP that cannot be avoided by switching RL algorithms—planning is conducted in the primitive action space rather than the abstract action space, thereby necessitating the significance of each primitive action. Future research could investigate modifications to Thinker that would allow model interaction through abstract actions, thereby allowing high-level and long-term planning.

Interestingly, the actor-critic applied to the raw MDP occasionally outperforms the one applied to the Thinker-augmented MDP. This could be attributed to the fact that we provide the agent with the model's hidden state instead of the true or predicted frames. It is conceivable that the actual frame is simpler to learn from, given that the model learns the encoding of frames by minimizing supervised learning loss rather than maximizing the agent's returns. Future studies could consider allowing the agent to directly observe the true and predicted frame, thus ensuring that the agent possesses a strictly greater capability than an agent on the raw MDP. Another factor could be the impact of unoptimized hyperparameters. As we increased the network's depth while maintaining the same learning rate during the transition from Sokoban to the Atari environment, the training process became more unstable, as demonstrated in some learning curves. It is plausible that the poor performance in certain games stems from these unoptimized hyperparameters.

Finally, as evidenced by the individual learning curves illustrated in Figure 11, and the median human-normalized score in Figure 9, we observe that the slope of the learning curve is generally steeper at the end of training compared to other baselines. We posit that mastering the use of a model represents an additional skill to be acquired, as opposed to an agent acting on a raw MDP. This necessitates extended training to learn this skill, however, once this skill is acquired, the final performance generally surpasses that of an agent on the raw MDP. As such, future work could experiment with a larger training budget than 200 million frames.

Table 4: Hyperparameters used in experiments.

| Parameter | Value |
|---|---|
| **Thinker-augmented MDP** | |
| Stage length $K$ | 20 |
| Maximum search depth $L$ | 5 |
| Augmented discount rate $\tilde{\gamma}$ | $\gamma^{\frac{1}{K}}$ |
| Planning reward scaling $c^p$ | Anneal linearly from 1 to 0 |
| | |
| **Model** | |
| Learning rate | Anneal linearly from 1e-4 to 0 |
| Optimizer | Adam |
| Adam beta | (0.9, 0.999) |
| Adam epsilon | 1e-8 |
| Replay buffer size | 200,000 |
| Minimum replay buffer size for sampling | 200,000 |
| Replay ratio | 6 |
| Batch size | 128 |
| Model unroll length $L$ | 5 |
| Reward loss scaling $c^r$ | 1 |
| Termination indicator loss scaling $c^d$ | 1 |
| Feature loss scaling $c^s$ | 10 |
| Value loss scaling $c^v$ | 0.25 |
| Policy loss scaling $c^\pi$ | 0.5 |
| Target for policy | Action distribution |
| | |
| **Actor-critic** | |
| Learning rate | Anneal linearly from 6e-4 to 0 |
| Optimizer | Adam |
| Adam beta | (0.9, 0.999) |
| Adam epsilon | 1e-8 |
| Batch size | 16 |
| Actor-critic unroll length | $20K + 1$ |
| Baseline scaling | 0.5 |
| Clip global gradient norm | 1200 |
| Entropy regularizer for real actions | 1e-3 |
| Entropy regularizer for imaginary and reset actions | 5e-5 |
| | |
| **Environment specific—Sokoban** | |
| Discount rate $\gamma$ | 0.97 |
| Input shape | (3, 80, 80) |
| | |
| **Environment specific—Atari 2600** | |
| Discount rate $\gamma$ | 0.99 |
| Grayscaling | No |
| Action repetitions | 4 |
| Max-pool over last N action repeat frames | 2 |
| Input shape | (12, 80, 80) |
| Frame stacking | 4 |
| End of the episode when life lost | Yes |
| Reward clipping | [-1, 1] |
| Sticky action | No |

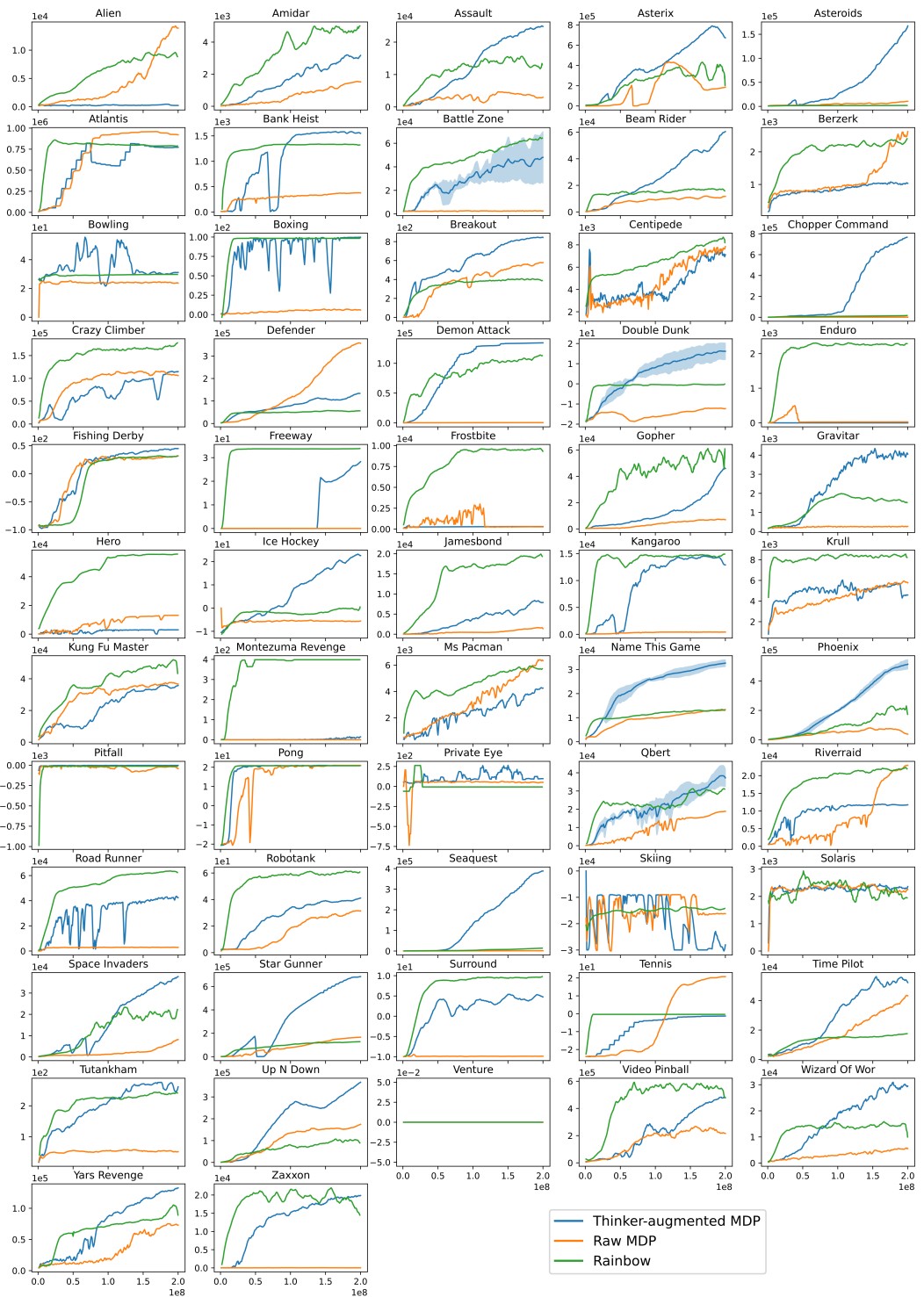

Figure 11: Learning curves for Thinker and baselines in Atari games. For all methods except Rainbow, each curve is computed as the running average of episode returns over the last 500 episodes. For the five selected Atari games, the shaded area represents the standard deviation across three seeds.

Table 5: Atari scores after 200M frames of training. Up to 30 no-ops at the beginning of each episode.

| | Rainbow | Raw MDP | Thinker Base Policy | Thinker Agent Policy |
|---|---|---|---|---|
| Alien | 9491.70 | **13868.80** | 229.20 | 230.10 |
| Amidar | **5131.20** | 1121.56 | 1338.75 | 3591.84 |
| Assault | 14198.50 | 6998.29 | 18856.51 | **24262.47** |
| Asterix | **428200.30** | 279440.00 | 238992.50 | 85254.00 |
| Asteroids | 2712.80 | 12570.40 | 40030.60 | **212805.50** |
| Atlantis | 826659.50 | **896017.00** | 808384.00 | 809160.00 |
| Bank Heist | 1358.00 | 378.50 | **1531.20** | 1528.90 |
| Battle Zone | **62010.00** | 2320.00 | 47553.33 | 48103.33 |
| Beam Rider | 16850.20 | 24202.46 | 15213.50 | **59533.14** |
| Berzerk | 2545.60 | **2664.00** | 1024.60 | 1038.30 |
| Bowling | 30.00 | 23.78 | **31.26** | 31.14 |
| Boxing | 99.60 | 8.17 | 99.51 | **99.86** |
| Breakout | 417.50 | 593.83 | 660.84 | **840.14** |
| Centipede | 8167.30 | **8201.71** | 5169.38 | 6507.05 |
| Chopper Command | 16654.00 | 1046.00 | 43691.00 | **843973.00** |
| Crazy Climber | **168788.50** | 97824.00 | 98755.00 | 116193.00 |
| Defender | 55105.00 | **385397.50** | 71378.50 | 123064.50 |
| Demon Attack | 111185.20 | 502.30 | 133410.20 | **135040.10** |
| Double Dunk | -0.30 | -1.72 | 15.27 | **16.09** |
| Enduro | **2125.90** | 26.03 | 0.00 | 0.00 |
| Fishing Derby | 31.30 | 32.46 | 38.58 | **43.68** |
| Freeway | **34.00** | 0.00 | 28.04 | 28.65 |
| Frostbite | **9590.50** | 269.70 | 303.30 | 303.10 |
| Gopher | **70354.60** | 5049.40 | 15871.40 | 55911.00 |
| Gravitar | 1419.30 | 267.00 | 3766.00 | **4226.00** |
| Hero | **55887.40** | 13117.30 | 2984.30 | 2987.80 |
| Ice Hockey | 1.10 | -5.27 | 20.62 | **22.99** |
| Jamesbond | N.A. | 1523.00 | 4171.50 | **7612.50** |
| Kangaroo | **14637.50** | 390.00 | 12424.00 | 13218.00 |
| Krull | **8741.50** | 5767.00 | 4663.10 | 4557.60 |
| Kung Fu Master | **52181.00** | 35209.00 | 32639.00 | 36391.00 |
| Montezuma Revenge | **384.00** | 0.00 | 12.00 | 14.00 |
| Ms Pacman | 5380.40 | **7117.66** | 2685.40 | 4206.10 |
| Name This Game | 13136.00 | 13391.00 | 25131.43 | **33085.20** |
| Phoenix | 108528.60 | 18201.70 | 156647.23 | **552719.37** |
| Pitfall | **0.00** | -38.17 | -2.95 | -2.51 |
| Pong | 20.90 | **20.97** | 20.96 | 20.84 |
| Private Eye | **4234.00** | 48.41 | 93.61 | 94.00 |
| Qbert | 33817.50 | 18491.75 | 20184.58 | **34849.33** |
| Riverraid | N.A. | **23983.60** | 11664.30 | 11696.00 |
| Road Runner | **62041.00** | 2988.00 | 37967.00 | 41950.00 |
| Robotank | **61.40** | 31.73 | 40.89 | 40.53 |
| Seaquest | 15898.90 | 1501.40 | 32992.90 | **432855.00** |
| Skiing | **-12957.80** | -16076.10 | -21515.92 | -24218.14 |
| Solaris | **3560.30** | 2136.60 | 2427.00 | 2323.40 |
| Space Invaders | 18789.00 | 8423.85 | 12880.40 | **42486.95** |
| Star Gunner | 127029.00 | 183274.00 | 290441.00 | **692283.00** |
| Surround | **9.70** | -9.86 | 1.89 | 4.28 |
| Tennis | 0.00 | **20.51** | -1.25 | -1.19 |
| Time Pilot | 12926.00 | 42851.00 | 37310.00 | **50767.00** |
| Tutankham | 241.00 | 51.60 | 262.00 | **268.14** |
| Up N Down | N.A. | 319317.80 | 337508.10 | **404120.10** |
| Venture | **5.50** | 0.00 | 0.00 | 0.00 |
| Video Pinball | **533936.50** | 281210.04 | 444212.40 | 512567.38 |
| Wizard Of Wor | 17862.50 | 6010.00 | 19304.00 | **31691.00** |
| Yars Revenge | 102557.00 | 72845.97 | 70000.15 | **124342.12** |
| Zaxxon | **22209.50** | 15.00 | 17716.00 | 19979.00 |

# E    Ablation Analysis

In this section, we investigate the significance of various elements of the Thinker-augmented MDP. All ablation experiments are performed in Sokoban under identical settings as those of the primary experiments, with results shown as the solving rate over the last 200 episodes.

**Hints and memory**    We examine the effect of removing all hints and other auxiliary statistics from the tree representation. Specifically, we remove the following statistics:

$$\{g^{\max}_{\text{child(root)}}, g^{\text{mean}}_{\text{child(root)}}, n_{\text{child(root)}}, g^{\max}_{\text{child(cur)}}, g^{\text{mean}}_{\text{child(cur)}}, n_{\text{child(cur)}}, g_{\text{root,cur}}, g_{\text{root,cur}} - \gamma^{d+1}\hat{v}_{\text{cur}}, g^{\max}_{\text{root}}\}$$

from the augmented state. These statistics are deemed auxiliary since, theoretically, the agent can compute these from the remaining statistics in the previous and current steps. The result of removing the hints is shown in Figure 12.

The removal of hints causes the agent to learn at a slower pace while slightly reducing the final performance. We theorize that the agent requires more training to learn to summarize the rollouts across multiple steps, leading to slower learning. With prolonged training, it could be feasible for the agent to learn this summarization and achieve the same final performance, thereby rendering the provision of these handcrafted hints unnecessary.

A specialized RNN architecture is employed to endow the agents with long-term memories while preserving training stability, as elaborated in Appendix C. We then consider the impact of replacing this RNN with a three-layer MLP with 200 hidden units, effectively removing the agents' memory. In the MLP, we add a skip connection from the input to each hidden layer, to ensure easier flow of information. The result of this adjustment is presented in the same figure. Interestingly, the removal of memory does not hamper performance in any notable way.

We hypothesize that as we have already summarized the rollouts for the agent by providing the hints, the agent is not required to perform this summarization itself and hence, does not require memory. To verify this hypothesis, we consider the scenario of removing both memory and hint, and the outcome is demonstrated in the same figure. The effect of eliminating memory proves to be highly significant when the hints are not provided. Consequently, we conclude that the presence of either the hints *or* memory is necessary for learning to plan.

**Base policy and imaginary actions**    We evaluate the effects of removing the base policy and its value. This leads to the tree representation being mostly empty as a number of statistics, such as the hints, are computed based on the node's value. Nonetheless, the agent still receives the model's hidden state as input. The result of this removal is shown in Figure 13.

The result shows that the agent fails to learn in such a scenario, indicating that the base policy is a critical component for learning to plan. We offer two potential explanations for this observation. Firstly, predicted values provide an agent with the ability to evaluate a leaf node more easily. This node evaluation allows an agent to evaluate a rollout without simulating the rollout till the end of the episode, thus allowing more shallow search. Even though the predicted value could theoretically be computed based on the hidden state of the node, which is provided to the agent, it is likely that this prediction is too difficult for the agent to learn by itself. Secondly, providing the base policy allows the agent to continuously improve upon the current policy. The base policy acts as a 'distilled' policy without planning, and the search corresponds to a learned policy improvement operator applied to this distilled policy. With the base policy and its value as inputs, the agent can learn different search algorithms as this policy improvement operator, completing the cycle of generalized policy iteration.

In addition, we consider the scenario where all imaginary actions are replaced with random actions, and we do not train the RL agent on these imaginary actions. The results are shown in Figure 13. This is intended to evaluate the benefits of learning imaginary actions. We find that performance significantly deteriorates, suggesting that the imaginary actions play a pivotal role in guiding real actions, consistent with the behavioural analysis in Appendix F.

**Maximum search depth**    We evaluate the effect of increasing the maximum search depth and the model unroll length $L$ from 5 to 10, thereby allowing the agents to search deeper. The result is shown in Figure 14. The initial learning speed is significantly slower, presumably due to the added time required for the model to learn to unroll over an increased number of steps. However, the final performance remains largely unaffected. This suggests that a deeper search does not necessarily lead

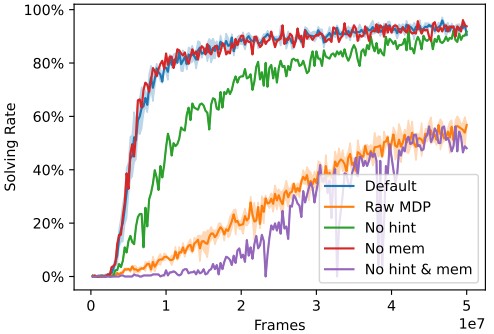

Figure 12: Ablation analysis on hints and memory.

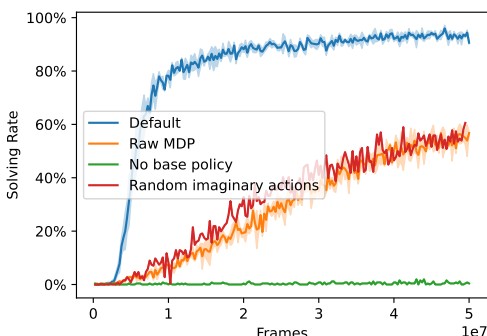

Figure 13: Ablation analysis on base policy and imaginary actions.

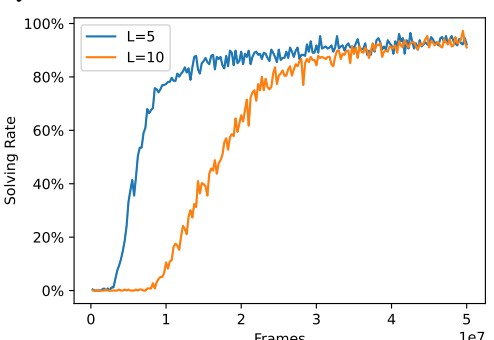

Figure 14: Ablation analysis on maximum search depth or model unroll length $L$.

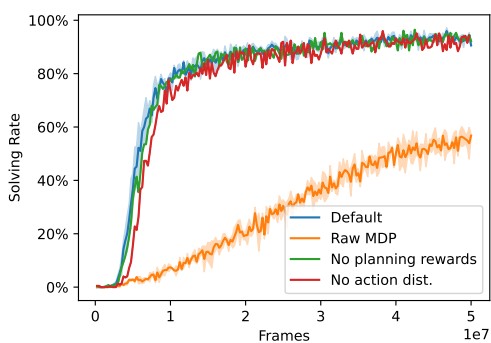

Figure 15: Ablation analysis on planning reward and action target.

to improved final performance. This finding aligns with previous studies indicating that a shallow search suffices to obtain satisfactory outcomes in MuZero [43].

**Planning rewards and action target**  Lastly, we discuss some non-essential components of the algorithm. To mitigate the sparse rewards in the augmented MDP, we added a heuristic reward, which we refer to as the planning reward, during the imaginary steps. The details of the planning reward are discussed in Appendix G. Figure 13 shows the results of excluding planning rewards. The removal of the planning rewards does not have any significant impact on the performance. This suggests that the planning reward may not be a critical component for learning to plan. Additional behavior analysis of a trained agent in Appendix F demonstrates that an agent trained without planning rewards seems to learn a similar planning behavior to that of an agent trained with planning rewards.

We used the action distribution instead of the sampled action for training the base policy in the model. However, this approach necessitates that the augmented MDP receive the action distribution from the agent, a procedure that is not consistent with the definition of an MDP. We alternatively experimented with using the sampled action as the target for training the base policy in the model, with the results depicted in Figure 13. This modification did not significantly impact overall performance, though it led to a slightly slower initial learning. Nonetheless, we recommend employing the action distribution in lieu of the sampled action, as the former encompasses more information and has a lower variance. The adherence to the conventional definition of an MDP, while theoretically pertinent, is of marginal consequence in practical applications.

## F   Agent Behavior Analysis

In this section, we delve into the behavior learned by the trained agents in the Thinker-augmented MDP of Sokoban. Our inquiry aims to understand the nature of the search policies acquired by the agents. Due to the use of the dual network, it is easy to visualize what the agent is 'planning' by plotting the predicted states from the model on each imaginary step. Figure 1 shows a typical stage, where the agent plans different ways of solving the task. Videos of the whole game and other Atari games are available at `https://youtu.be/0AfZh5SR7Fk`.

Though the video is valuable in discerning the plan of an agent, a more quantitative analysis is warranted. Does the agent adhere to handcrafted planning algorithms, such as the $n$-step exhaustive search or MCTS, or does it learn novel planning algorithms? Given that the agents learn three types of actions—reset, imaginary, and real actions—we will analyze these actions individually.

**Reset Action**   In a $n$-step exhaustive search, which involves iterating through every possible multi-step action sequence and choosing the one with the highest rollout return, a reset is triggered when the search depth hits $n$. Similarly, MCTS initiates a reset at a leaf node. However, the trained agent seems to learn different behavior of resetting.

As visualized in the video, the agents often chose to reset when envisioning a poor rollout, for instance, when imagining a box being moved to an irreversible state. To illustrate this further, we can examine the distribution of Temporal Difference (TD) errors of the rollout from the root node to a node $i_n$, defined as:

$$\delta_{i_n} := \hat{r}_{i_1} + \gamma^2 \hat{r}_{i_2} + ... \gamma^n \hat{r}_{i_n} + \gamma^{n+1} \hat{v}_{i_n} - \hat{v}_{\text{root}}, \tag{24}$$

where $\{\text{root}, i_1, i_2, ..., i_n\}$ denotes the path from the root node to the node $i_n$. This TD error can be interpreted as the quality of the rollout relative to the current policy. We compare the distribution of TD errors of the current rollout $\delta_{\text{cur}}$ where the agent decided to reset with those where it did not, and the result is visualized in the left and middle panels of Figure 16.

We observe that the TD error distribution when agents opt to reset tends to lean more towards the left as compared to when they choose not to reset. This indicates a higher propensity for the agent to reset when the quality of the rollout is inferior. The thin tail on the left side of the distribution represents instances where the agent envisions a poor plan, such as reaching an irreversible state. Under such circumstances, the agent exhibits a high probability of resetting. This suggests that the agents learned to factor in the current rollout's quality in their decision to reset. This approach diverges from planning strategies like MCTS, which mandates a reset upon reaching a leaf node.

An additional dimension worth investigating is the *expansion length*, defined as the number of steps the agent executes beyond the last expanded node. In the case of MCTS, this length is consistently one as the algorithm requires a reset at every leaf node. However, as shown in the right panel of Figure 16, the trained agents generally exhibit a longer expansion length. In fact, when the maximum search depth $L$ is increased to 10, the agent learns to search much deeper, as depicted in the same figure.

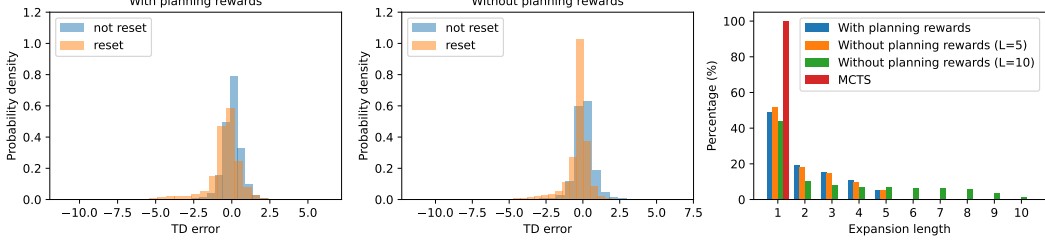

Figure 16: Left and Middle: Probability density of TD error upon choosing to reset and not to reset. These statistics are computed for agents trained with (Left) and without planning rewards (Middle). Right: Percentage of expansion length for agents trained with and without planning rewards at different maximum search depths $L$. All statistics are averaged over 20000 stages, each starting from random initial states.

Coupled with the aforementioned insights into the TD error distribution, we see that the agents prefer to persist with their existing plan unless they encounter an unfavorable leaf node or reach the set maximum search depth. These observations suggest that the agents tend to formulate plans that are both deep and narrow. Arguably, these learned behaviors mimic our planning processes more closely—we typically continue to think about a good plan, rather than simultaneously considering multiple plans and expanding each by a single step.

**Imaginary Action**   In a $n$-step exhaustive search, imaginary action is chosen sequentially, while in MCTS, imaginary action is chosen according to the upper confidence bound (UCB). As for the trained agents, it is difficult to deduce the formula used to select the imaginary action as it is the output of a deep neural network. However, the maximum TD errors in a stage, defined as $\delta^{\max} := \max_k \delta_k$ where $1 \leq k < K$ denotes the step within a stage, could be used as a proxy to evaluate the effectiveness of the search[4]. The distribution of maximum TD errors of search using the trained agent, UCB, and random search is shown in Figure 17. Note that we use the learned model of the trained agent when applying UCB.

We observe that the distribution of maximum TD errors has a slightly heavier tail on the right side when searching with a trained agent. This can be better visualized in log scale as shown in Figure 18. This tail mostly corresponds to the case of solving or being closer to solving a level in the imagination where the current policy has a high chance of not solving it. This scenario is relatively rare given that the current policy is capable of solving approximately 95% of levels, which explains the thin tails in all cases. Thus, the heavier tail suggests that the agents learn to search more effectively than MCTS within the same search budgets. This finding is consistent with previous studies on replacing MCTS with neural networks in supervised learning settings [19].

We conjecture that the difference between an RL agent and MCTS in this distribution would be much larger if the model were also trained with MCTS, given the large number of simulations required for MCTS to achieve good performance in Sokoban [19]. It is also noteworthy that even without planning rewards, the agent is still able to collect large maximum TD errors, which implies that the agent can learn to search effectively without the guidance of heuristics.

**Real Action**   In a $n$-step exhaustive search, the real action is taken as the child node with the highest rollout return, while in MCTS, the real action is usually selected based on the visit count of each child node. In contrast, our trained agents appear to learn to use a combination of both.

To further investigate how real action is selected given the same search result, we used the imaginary action and reset action outputs from the trained agent and considered two different ways of selecting the real action: (i) the real action output by the trained agent, (ii) the real action output by MCTS, which is sampled based on visit counts with different temperatures $T \in \{0.25, 1\}$ as in MuZero [5]. For each method, we calculated the portion of the real action that corresponded to the highest visit count, the second highest visit count, and so on. The result is shown in the left panel of Figure 19. Additionally, we computed the same for the mean rollout return and the maximum rollout return of child nodes, as shown in the middle and right panels of the figure.

The figure shows that most of the real action selected by the agents corresponds to the child node with the highest visit counts or rollout returns. The figures also suggest that the maximum rollout return is the most indicative of the agent's real action among the three statistics. In contrast, MCTS tends to select the real action with the highest visit count. Notably, the action with the highest visit count does not necessarily correspond to the action with the highest rollout return, as evidenced by MCTS with $T = 0.25$, which chooses the action with the highest visit count with over 80% probability, yet only around half of these actions correspond to the one with the highest rollout return. The simultaneous consideration of both rollout returns and visit counts by the trained agent seems more intuitively reasonable—it is unwise to follow a poor plan even if it is simulated numerous times.

**Comparison with MCTS**   Lastly, to compare the efficiency of the trained agents across all three action sets with MCTS, we assess the solving rates of both the trained agents and MCTS. In MCTS, we utilize the model learned by the agent at the end of training, and we test a range of simulation

---

[4]This maximum TD error is proportional to the undiscounted sum of planning rewards in a stage.

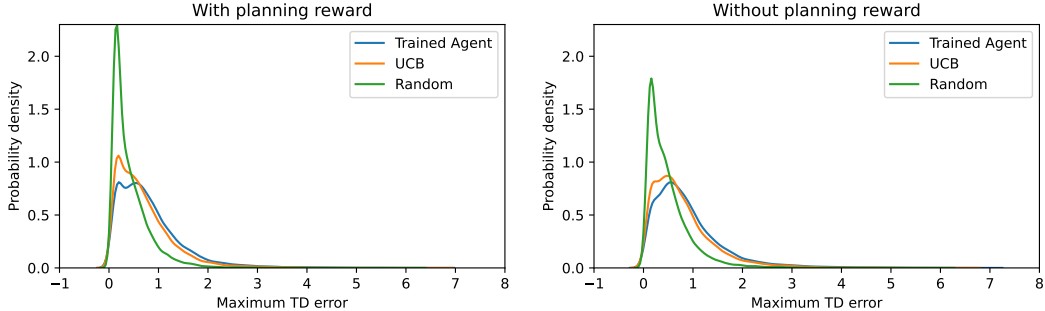

Figure 17: Maximum TD error collected by different search methods, using the model learned by the agent. All statistics are averaged over 20000 stages, each starting from random initial states.

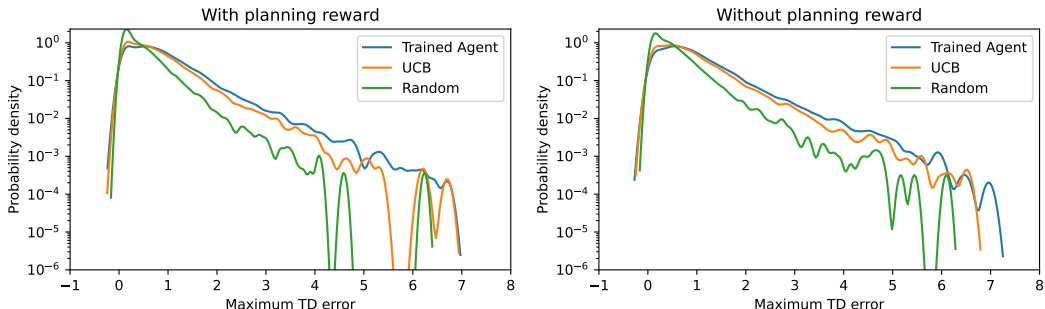

Figure 18: Maximum TD error collected by different search methods in log scale, using the model learned by the agent. All statistics are averaged over 20000 stages, each starting from random initial states.

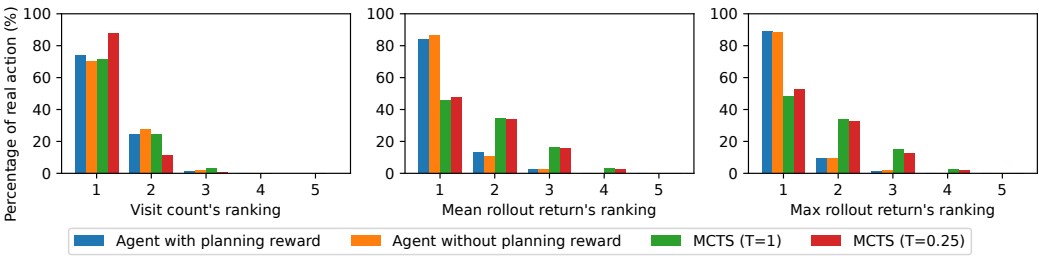

Figure 19: Percentage of chosen real action grouped by the ranking of visit count, mean rollout return, and maximum rollout return. All statistics are averaged over 20000 stages, each starting from random initial states.

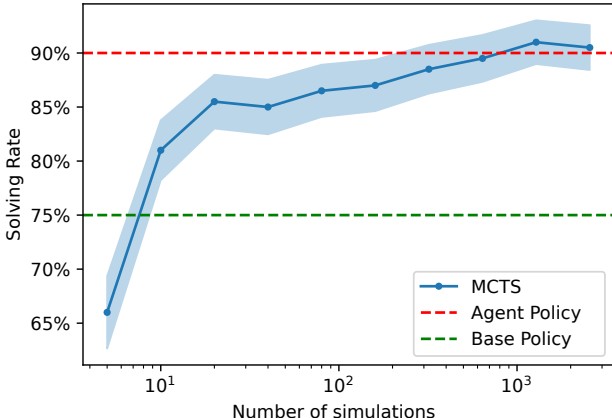

Figure 20: Final solving rate on Sokoban training levels as achieved by MCTS with varying numbers of stage length, compared to the performance of the trained agent policy and the base policy.

steps[5] spanning over {5,10,20,40,80,160,320,640,1280,2560}. The results are depicted in Figure 20. We observe that MCTS needs 1280 simulation steps to match the performance of a trained agent, which achieves the same results with only 20 stage length. This notable difference highlights the potential benefits of trained agents compared to handcrafted algorithms—they exhibit the capacity to conduct searches with increased efficiency, particularly within limited computational resources.

In conclusion, the analysis in this section suggests that the trained agents appear to learn a planning algorithm that differs from common handcrafted planning algorithms. These agents learn to plan deeply and reset upon encountering a poor node, while being able to search for a good plan within a limited number of model simulations. It is likely that the agents also take advantage of the model's hidden state, which contains rich information about the root and predicted states, when deciding upon the three types of actions. However, a comprehensive investigation of this particular dynamic lies beyond the scope of our current analysis.

---

[5]Note that the stage length can be smaller than the number of simulations, as the former includes traversing expanded nodes while the latter does not.

# G    Planning Rewards

In the main paper, we use the extended real reward as the augmented reward:

$$\tilde{r}_k := \begin{cases} 0 & \text{if } k \leq K, \\ r, & \text{if } k = K + 1, \end{cases} \tag{25}$$

where $r$ denotes the real reward obtained from the real step. However, this sparse reward may make learning difficult. We explore adding an auxiliary reward, which we call the *planning reward*, to help guide the learning of imaginary and reset actions. We define the planning reward $\tilde{r}_k^p$ on step $k$ to be:

$$\tilde{r}_k^p := \begin{cases} c^p \left( g_{\text{root},k}^{\max} - g_{\text{root},k-1}^{\max} \right), & \text{if } k \leq K \\ 0, & \text{if } k = K + 1, \end{cases} \tag{26}$$

where $g_{\text{root},k}^{\max}$ denotes the maximum rollout return of the root node on step $k$ within a stage, as defined after Equation 1, and $c^p \geq 0$ is a scalar hyperparameter that controls the strength of the planning reward.

Note that the total undiscounted sum of planning rewards in a stage is:

$$\sum_{k=2}^{K+1} \tilde{r}_k^p = c^p(g_{\text{root},K}^{\max} - g_{\text{root},0}^{\max}) \tag{27}$$

$$\propto \hat{r}_{i_1^*} + \gamma^2 \hat{r}_{i_2^*} + ...\gamma^n \hat{r}_{i_n^*} + \gamma^{n+1} \hat{v}_{i_n^*} - \hat{v}_{\text{root}}, \tag{28}$$

where $\{\text{root}, i_1^*, i_2^*, ..., i_n^*\}$ refers to the rollout with the highest return, $g_{\text{root},i}$, in the entire stage. As $\hat{v}_{\text{root}}$ is not influenced by imaginary and reset actions, the return of a stage is equivalent to $\hat{r}_{i_1^*} + \gamma^2 \hat{r}_{i_2^*} + ...\gamma^n \hat{r}_{i_n^*} + \gamma^{n+1} \hat{v}_{i_n^*}$, the maximum rollout return. Consequently, the planning reward is a heuristic directing the agent to maximize the maximum rollout return, while not penalizing for searching paths with a low rollout return to encourage exploration in imagination.

However, this planning reward should not be given to real actions to prevent incentive misalignment. For example, collecting a large amount of planning rewards is possible by continually doing nothing in real steps but imagining a good plan. As such, the planning reward signal has to be separated from the real reward signal $\tilde{r}_k$ instead of being added together. We also decay $c^p$ from 1 to 0 throughout training to further ensure that the imaginary action is aligned with the real reward function. Combined, the augmented reward contains both the real reward and the planning reward: $r^{\text{aug}} := (\tilde{r}, \tilde{r}^p)$.

The incorporation of planning rewards necessitates certain modifications to the actor-critic algorithm in our experiments. Specifically, we separately learn the value of these two distinct rewards when training the critic, resulting in two different TD errors corresponding to the real and the planning rewards. In training the critic for the planning reward, we treat each stage as a single episode. This approach ensures that the planning reward incentives the agent to optimize the maximum rollout returns exclusively for the current stage. Furthermore, when using the TD errors to train the actor, we impose a mask on the planning rewards' TD error, setting it to zero during a real step.

Experiments in Appendix E show that planning rewards only provided a minimal increase in initial learning speed, indicating that the real rewards are sufficient for learning to plan. We conjecture that sparse real rewards are not a problem, as the TD errors induced by real rewards may not be sparse. For example, if the critic learns to use the maximum rollout return $g_{\text{root},k}^{\max}$ as the estimated value, which is provided in the tree representation, then the TD errors in an imaginary step equal the planning rewards, except for the adjustment due to the discount rate.

# H Relationship with Generalized Policy Iteration and Meta-Reinforcement Learning

## H.1 Generalized Policy Iteration Algorithm

Applying an RL algorithm to the Thinker-augmented MDP can be viewed as an instance of a generalized policy iteration algorithm [29, 30]. Policy improvements are carried out by an agent that conducts searches within the model. Given the base policy, these searches yield an improved policy. Simultaneously, the policy evaluation is done by the learning of the value-policy network. This involves training the network on trajectories generated by the improved policy, so as to learn the values of this improved policy and project the improved policy back to the base policy. A noteworthy divergence from a typical generalized policy iteration algorithm is the learning of the policy improvement operator, instead of employing a handcrafted one such as MCTS.

Given this perspective, a crucial question arises: how effective is a learned policy improvement operator in comparison to a handcrafted one? To address this question, we evaluate the efficiency of different policy improvement operators, such as MCTS and an RL agent, during a single step of policy improvement.

Specifically, we consider the Thinker-augmented MDP with a fixed base policy $\hat{\pi}$ and its value $\hat{v}$. We do not update the base policy that is described in Section 4.4. The base policy is a given policy that performs slightly better than a random policy. For simplicity, we employ the true environment dynamics, represented as $(P, R)$, as the model in the augmented MDP and do not impose a limit on the maximum search depth, given that we are using the true environmental dynamics. Sokoban serves as the real environment in our experiments.

We experiment with three different types of policy improvement operators: (i) $n$-step exhaustive search, (ii) MCTS, and (iii) RL agents. In (iii), we trained the agent from scratch in the Thinker-augmented MDP. Only the tree representation $u_{\text{tree}}$ is provided as input to the agent, so the agent cannot learn to act based on the real states directly. The results are shown in Figure 21. Due to the low solving rate, we report the return of an episode here for better illustration. The maximum return in Sokoban is around 14, and a return of 4 gives at most 30% solving rate.

From the figure, we see that the base policy achieves a return of 0.12, and all policy improvement operators are able to improve on this policy and achieve a better return. We observe that the policy improvement gap for RL agents widens as the value of $K$ increases. With $K$=10, the agent's performance is similar to that of a 3-step exhaustive search, which requires 155 model simulations. Increasing $K$ to 20, the performance is slightly worse than a 4-step exhaustive search, which requires 780 model simulations. At $K$=40, the agent performance surpasses a 5-step exhaustive search, which requires 3905 model simulations. In addition, the agent at $K$=10 also shows comparable performance to MCTS that uses 60 model simulations. It is also worth noting that the number of model simulations used by the agent is typically less than $K$, given that imaginary steps also include the traversal through expanded nodes.

The results here demonstrate that increasing $K$ to 40 can still enhance performance, whereas in the main paper, the performance improvement gained by increasing $K$ above 10 is insignificant. We conjecture that once the value-policy network is trained, the model can learn to predict the values of leaf nodes more accurately, reflecting the value of the current policy instead of the base policy. This eliminates the need for a deep search to evaluate a rollout more precisely, rendering a large $K$ unnecessary.

The results also underscore the importance of a learned base policy. The agent achieves a maximum solving rate of less than 30%, whereas a 95% solving rate is attainable when the base policy undergoes training as well. This outcome is intuitive, considering that the agent gives an improved version of the base policy, and thus the agent policy will not significantly outperform it. If a fixed poor base policy is used, the agent policy will not perform well.

Nonetheless, these findings suggest that an agent can learn to perform policy improvement and achieve similar performance to handcrafted algorithms while requiring significantly fewer simulations. However, it requires a moderate number of transitions for an agent to learn this policy improvement operator.

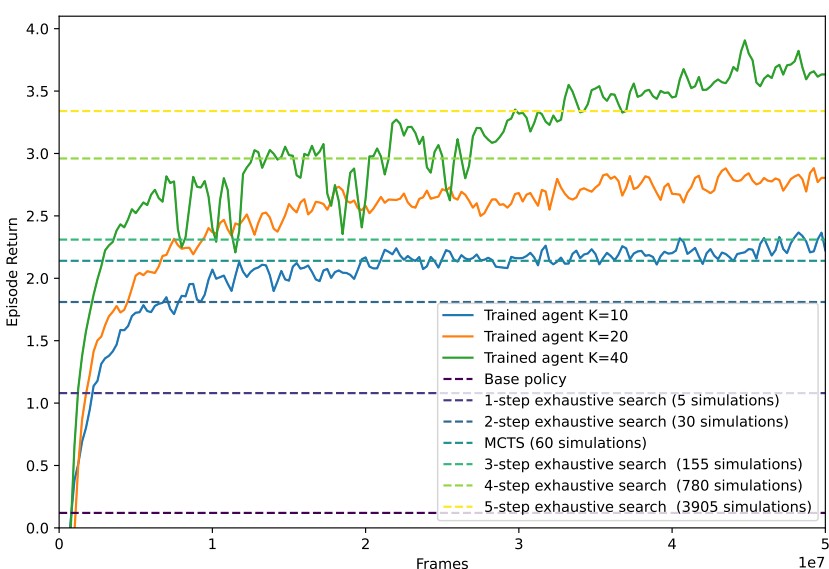

Figure 21: Comparison of different policy improvement operators applied to a fixed base policy in Sokoban. For trained agents, the running average of returns from the last 2000 episodes is shown, whereas for others, an average return from 2000 episodes is shown.

## H.2 Meta-Reinforcement Learning

In this section, we explore the relationship between meta-reinforcement learning (meta-RL) algorithms [48] and the Thinker algorithm. Meta-RL is a method designed to learn how to perform reinforcement learning. Its objective is to develop a policy that can quickly adapt to a variety of new tasks derived from a distribution of tasks, using minimal data. While meta-RL is often used in scenarios involving a task distribution, it is also applicable to single-task scenarios [49, 50], which is our primary focus here.

In the meta-RL framework, a trial typically comprises $L$ episodes. During a trial, the agents initially encounter $K$ episodes—termed *burn-in episodes*. Notably, the return within these episodes does not contribute to the main objective. Instead, the primary goal is to maximize the return of the remaining $L - K$ episodes, known as *evaluation episodes*. Consequently, the purpose of a meta-RL algorithm is to use the knowledge gained throughout the entire trial to maximize returns during the evaluation episodes. One common class of meta-RL algorithm applies an RL algorithm with an RNN that can access previous actions and rewards, and the hidden state is not reset across episodes. As such, the knowledge of the trial is embedded in the RNN's hidden states [27, 28].

If the model perfectly represents the true environment dynamics, we can align the Thinker-augmented MDP with meta-RL by considering imaginary rollouts as (truncated) burn-in episodes, and real transitions as the evaluation episode in meta-RL. As agents only aim to maximize the return in real transitions, the objectives of meta-RL and RL agents on the Thinker-augmented MDP coincide. Moreover, the purpose of burn-in episodes in meta-RL or imaginary rollouts in planning is the same – to guide better actions in the evaluation episodes or real transitions, possibly through risk-taking behavior in them.

However, unlike conventional meta-RL, which uses a fixed number of burn-in episodes followed by evaluation episodes, the Thinker-augmented MDP interleaves the burn-in episodes within a single evaluation episode. In other words, for each step in the evaluation episode with state $s_t$, we allow agents to have a few burn-in episodes starting from $s_t$.

In meta-RL, it is generally assumed that the burn-in episodes and evaluation episodes are drawn from the same MDP, but this assumption does not hold when the burn-in episodes are generated by a learned model. But if the MDP underlying the burn-in episodes, $\hat{\mathcal{M}}$, is similar to that of the evaluation episode, $\mathcal{M}$, then the meta-RL algorithm may also learn how to utilize the collected data while accounting for the difference. This is analogous to how the RL agent in the Thinker-augmented MDP learns to use the imaginary rollout for better action selection, despite the possible inaccuracies of the model.

In summary, there is a close relationship between learning to plan and learning to learn. The former employs a model on each step to decide better actions at that state, while the latter uses episodes within a trial to determine better actions in the evaluation episodes. This highlights the converging ideas in meta-learning and planning, both aiming to make efficient use of available information for decision-making.

# I Miscellaneous

## I.1 Related Work on the Dual Network

The dual network is similar to the model in MuZero [5], except for (i) the use of the dual network architecture and (ii) the prediction of future states trained by feature loss. Improvements to MuZero's model have been proposed. EfficientZero [51] deviates from MuZero by adding SimSiam loss [52] to match future representations. We briefly experimented with it, but the result was only slightly better than that of MuZero's model, possibly because this loss works better in a limited data regime where EfficientZero was designed and evaluated. [43] also proposes to predict future states as an auxiliary loss but does not use the dual network architecture. In our experiments, this addition led to better performance, as shown in Figure 4, but the result was still significantly worse than that of the dual network.

Most other types of models used in model-based RL do not predict values and policies, as future values and policies are often not required in the algorithm. Among these models, various loss functions have been proposed for training the model to acquire meaningful representations for the task. [53] employs a VAE [54] to encode the state while Dreamer [33] utilizes a sequential VAE. Given the extensive body of research in model-based RL, we direct readers to the review [55] for a detailed exploration of this field.

## I.2 Planning Capacity

In this section, we discuss the planning algorithms that can be implemented in the Thinker-augmented MDP. Uninformed search strategies [56, Ch 3.4] that do not require backward search, including Breadth-first search (BFS), Depth-first search (DFS), MCTS can be implemented by the RL agent in the Thinker-augmented MDP. For example, to implement BFS with two actions available, the agent can select $(a_1, \text{reset})$, $(a_2, \text{reset})$, $(a_1, \text{not reset})$, $(a_1, \text{reset})$, $(a_1, \text{not reset})$, $(a_2, \text{reset})$, ... as the imaginary actions. If we treat the values as heuristics, then informed search strategies [56, Ch 3.5] that do not require backward search, can also be implemented. For example, A* could be implemented by expanding the unexpanded node that has the highest rollout return in the tree, which can be achieved by keeping a record of unexpanded nodes along with the rollout return in the memories.

Nonetheless, as an MDP does not necessarily involve a goal state as in planning problems and our trained model does not support backward unrolling, any backward search from a goal state, such as bidirectional search [56, Ch 3.4] and partial-order planning [56, Ch 10.4], cannot be implemented by the agent. Agents are also unable to implement planning algorithms that involve heuristics beyond value function, such as GRAPHPLAN [56, Ch 10.3] that employs heuristics based on planning graphs. In summary, an RL agent under the Thinker-augmented MDP can implement any forward-based planning algorithms that do not involve heuristics beyond the value functions.

In practice, the learned planning algorithm is very different from the aforementioned handcrafted planning algorithms, as illustrated in Appendix F and the video visualization. This is because, unlike traditional planning algorithms whose goal is to solve the task in the search, the RL agent's primary incentive is to provide useful information for selecting the next immediate real action.

