# OpenReview forum: "Thinker: Learning to Plan and Act"
_NeurIPS.cc/2023/Conference — NeurIPS 2023 poster_

### Official Review · Reviewer_fWg3 · 2023-07-06

**Soundness:** 3 good
**Presentation:** 4 excellent
**Contribution:** 3 good
**Rating:** 8
**Confidence:** 3

**Summary:**

This paper presents *Thinker*, a method for augmenting an MDP with a learned model so that model-free algorithms can perform “planning” on the augmented MDP. Specifically, Thinker creates a cross-product MDP between the “real world” and a learned world model. This augmented MDP has a union action set that enables an RL agent to either take an action in the real world at a given step, or take an action in the world model. Decision-making is broken down into K-1 steps of acting in the world model (“planning”), followed by a single step of acting in the real model, potentially exploiting information from the K-1 steps of simulation. The authors argue that this allows model-free algorithms to learn how to execute various common planning algorithms.

Evaluation is performed on Sokoban and 57 ALE games. The authors find that IMPALA applied to the Thinker-augmented MDP achieves a higher solve rate for a fixed number of world steps than IMPALA applied to the base MDP, as well as better results than one pre-existing baseline. Analogous results hold on the ALE benchmarks.

**Strengths:**

- **[S1]** The fact that this method merely augments the MDP with some extra transitions (and a learned world model) means that it is both conceptually simple and broadly applicable to many existing reinforcement learning algorithms.

- **[S2]**  Experimental evaluation is thorough. Sokoban is a genuinely difficult benchmark for planning (it’s PSPACE hard in general), and the paper also evaluates across 57 Atari games, which greatly reduces the probability that results were cherry-picked. The ablations in the appendix were also quite thorough.

- **[S3]**  The paper is tackling a problem of significance for the community (planning in reinforcement learning) and was clearly written.

**Weaknesses:**

- **[W1]** It’s not clear how much the empirical benefits of Thinker come from actually doing search as opposed to just giving the model a more expressive feature space (since the policy can depend on the rollout history at each node), or something else of that nature. The fact that the algorithm’s benefits seem to plateau after K=10-20 planning steps is particularly surprising: traditional planning algorithms might take tens of thousands of node expansions to find a plan of tens to hundreds of steps.

  The ablations go some way towards resolving this issue, particularly the ablations that decrease K and the ablation that removes some of the redundant features from the state space. However, more analysis of what the algorithm is actually doing during the planning phase would be helpful. For instance, the paper could include a qualitative analysis of Sokoban rollouts that tries to explain what kind of planning the algorithm is doing. It could also include an extra ablation in which actions are randomly chosen during the planning phase, rather than being chosen by Impala—does learning the planner with Impala actually help?

- **[W2]** As I understand it, the main contribution of the paper is a way of augmenting MDPs, as opposed to a specific algorithm. Indeed, the paper uses this as a justification for the fact that it only has one third-party baseline for Sokoban and one for Atari. However, the paper only applies the Thinker augmentation to one RL algorithm (Impala). This is not a fatal problem for the paper, but it would be valuable to see the Thinker augmentation applied to other model-free RL algorithms to verify that it actually does work well for algorithms other than Impala.

- **[W3]** The paper appeals to the “generality” and “universality” of the Thinker augmentation, applying that it can express any (?) planning algorithm. I think this is over-claiming: the Gym-like interface available during planning steps makes it hard to implement even a simple algorithm like BFS, and I’m not sure how Thinker would express more complex algorithms like regression (backwards search from the goal) or partial-order planning, not to mention planning algorithms that require complicated heuristics. It would be helpful if the paper was more precise about what planning algorithms Thinker can and can’t express.

Note that W1 is my biggest objection, since I feel it cuts against the core claim of the paper. I would really like to see this addressed in the rebuttal. W3 is a more minor objection but is pretty easy to address, so it would be good to see the paper updated to reflect that (or to see an argument in the rebuttal for why it can’t be done). W2 is a big request for more experiments & so I understand if the authors don’t have time or compute to do this; it’s more of a nice-to-have than something I consider essential to publication.

I’m marking this a weak accept for now due to the issues listed above, but I’m overall pretty happy with the paper, and expect to upgrade my rating later so long as the rebuttal is reasonable & no other reviewers identify major flaws in the experiments or conceptual basis of the idea.


**Questions:**

1. Please explain in more detail why you believe the benefits of this algorithm come from it doing planning (systematic search), as opposed to some other benefit of the Thinker augmentation (i.e. address W1 above).
2. Is there a crisp way of separating the class of planning algorithms that Thinker can emulate from those that it can’t? (W3)

**Limitations:**

I felt the limitations were pretty thorough, although I would have liked it if this section listed the expressiveness concern (W3) as well, or at least made more precise claims about the expressiveness of Thinker-augmented model-free RL algorithms.

---

> ### Author Rebuttal · Authors · 2023-08-09
>
> We thank the reviewer for the constructive comments and detailed feedback! We appreciate the reviewer’s concise summary of the proposed algorithm and acknowledgement of our experiments. The weaknesses are addressed as follows:
>
> **(R1) Algorithm’s benefits seem to plateau after K=10-20 planning steps is particularly surprising**
>
> Please refer to (G3) in the global response for the ablation results on imaginary actions and section G in the appendix for the analysis of learned agents’ behaviour. We found that RL agents can learn to search much more efficiently compared to other handcrafted planning algorithms, as shown in Fig G5 in the appendix. [1] also shows that a trained neural network can search much more efficiently than MCTS in a supervised learning setting.
>
> Note that if the better performance of Thinker stems merely from the expanded feature spaces of the model, and the imaginary actions from the RL agent don't enhance performance, then one would expect the rollouts in runs without planning rewards to be completely random. This is because, if the planning reward is disabled, the RL agent would only be incentivized to maximize true returns, and the entropy regularization would push all non-useful actions towards a uniform distribution. However, we observe that the rollouts in runs without planning rewards remain specific and meaningful (e.g., at 8:35 timestamp of the video in the paper, the agent imagines solving the task in various ways). The new experimental results in Fig A3 provide more direct evidence of the benefits of the learned imaginary actions.
>
> **(R2) Thinker augmentation applied to other model-free RL algorithms**
>
> We aimed to demonstrate the general improvement offered by the augmented MDP by running a standard actor-critic algorithm across a wide range of domains (56 different environments). While it would be interesting to explore how the augmented MDP interacts with various RL algorithms, rather than different environments, the substantial computational cost associated with the algorithm prohibits us from doing so. We leave this for future research. Moreover, given that we employed a straightforward actor-critic algorithm without modification, it's reasonable to expect that the augmented MDP would be compatible with more advanced RL algorithms. We greatly appreciate the reviewer’s understanding on this matter.
>
> **(R3) Planning capacity under the augmented MDP**
>
> We recognize that our claim that an RL agent can express most planning algorithms is misleading, and we apologize for it. We will revise the paper (e.g., replace line 30 *any planning algorithm* with *common forward-based planning algorithms*)  and include the following discussions in the appendix:
>
> *An RL agent under the Thinker-augmented MDP can implement any forward-based planning algorithms that do not involve heuristics beyond the value functions. Uninformed search strategies [2, Ch3.4] that do not require backward search, including Breadth-first search (BFS), Depth-first search (DFS), MCTS can be implemented by the RL agent. For example, to implement BFS with two actions available, the agent can select (action_1, reset),  (action_2, reset), (action_1, not reset), (action_1, reset), (action_1, not reset), (action_2, reset), … as the imaginary actions\*. If we treat the values as heuristics, then informed search strategies [2, Ch3.5] that do not require backward search, can also be implemented. For example, A\* could be implemented by expanding the unexpanded node that has the highest rollout return in the tree, which can be achieved by keeping a record of unexpanded nodes along with the rollout return in the memories.*
>
> *Nonetheless, as an MDP does not necessarily involve a goal state as in planning problems and our trained model does not support backward unrolling, any backward search from a goal state, such as bidirectional search [2, Ch3.4] and partial-order planning [2, Ch10.4], cannot be implemented by the agent. Agents are also unable to implement planning algorithms that involve heuristics beyond value function, such as GRAPHPLAN [2, Ch10.3] that employs heuristics based on planning graphs.*
>
> *In practice, the learned planning algorithm is very different from the aforementioned handcrafted planning algorithms, as illustrated in Appendix G and the video visualization. This is because, unlike planning algorithms whose goal is to solve the task in the search, the RL agent's primary incentive is to provide useful information for selecting the next immediate real action.*
>
> \*We note that searching in this manner is not efficient, as the agent must begin expanding from the root node every time instead of directly starting from the unexpanded nodes (or the *frontier nodes*). Also, implementing BFS requires the agent to have memory, as whether one should reset at a specific node depends on whether all nodes at the same level have been expanded, which is beyond the auxiliary statistics.
>
> The questions raised are addressed as follows:
>
> > Please explain in more detail why you believe the benefits of this algorithm come from it doing planning (systematic search)...
>
> Please refer to our answer (R1) above.
>
> > Is there a crisp way of separating the class of planning algorithms that Thinker can emulate from those that it can’t?
>
> Please refer to our answer (R3) above.
>
> We hope that the answers provided above adequately address the issues raised by the reviewer, and we kindly ask the reviewer to consider adjusting the score based on our responses. We are grateful for the reviewer’s comments and welcome any further questions!
>
> **References**
>
> [1] Guez, A., Weber, T., Antonoglou, I., Simonyan, K., Vinyals, O., Wierstra, D., ... & Silver, D. (2018, July). Learning to search with MCTSnets. In International conference on machine learning (pp. 1822-1831). PMLR.
>
> [2] Stuart, R., & Peter, N. (2010). Artificial Intelligence A Modern Approach Third Edition.

---

> > ### Comment · Reviewer_fWg3 · 2023-08-18
> > **Good rebuttal**
> >
> > ### Overall take
> >
> > * The authors have addressed all concerns that I thought needed to be addressed before publication.
> > * I'm in favor of acceptance and have updated my score.
> > * I've read the other reviews and they didn't give me reason to change my score much either way given the author rebuttal.
> >
> > ### (R1) Algorithm’s benefits seem to plateau after K=10-20 planning steps is particularly surprising
> >
> > > Note that if the better performance of Thinker stems merely from the expanded feature spaces of the model, ... then one would expect the rollouts in runs without planning rewards to be completely random [because] entropy regularization would push all non-useful actions towards a uniform distribution.
> >
> > This is a good point that I hadn't considered.
> >
> > > The new experimental results in Fig A3 provide more direct evidence of the benefits of the learned imaginary actions.
> >
> > Thank you! A3 resolves my main concern here.
> >
> > ### (R2) Thinker augmentation applied to other model-free RL algorithms
> >
> > > the substantial computational cost associated with the algorithm prohibits us from doing so
> >
> > Fair enough. The existing experiments look pretty brutal, computationally. Well done for getting through all of them!
> >
> > ### (R3) Planning capacity under the augmented MDP
> >
> > > We will revise the paper (e.g., replace line 30 any planning algorithm with common forward-based planning algorithms) and include the following discussions in the appendix: ...
> >
> > Thanks a lot, this is very precise and I think strengthens the paper.

---

### Official Review · Reviewer_9fjR · 2023-07-07

**Soundness:** 3 good
**Presentation:** 1 poor
**Contribution:** 3 good
**Rating:** 6
**Confidence:** 4

**Summary:**

The authors propose Thinker, a model-based RL algorithm that transforms a MDP into one where at each step, the agent is required to generate plans in imagination before selecting a real world action. Then through the RL gradient, the agent learns to plan in order to act. The authors get good results in Atari and Sokoban benchmarks.

**Strengths:**

- An interesting way of combining planning and learning by integrating planning into the MDP itself. By viewing planning as part of the MDP itself, an RL agent maximizing this MDP will learn how to plan rather than relying on a given planning algorithm to maximize reward.
- Good results on Sokoban, a hard planning task, and analysis (some issues, see below)


**Weaknesses:**

There are two major concerns I have, one in experimentation and one in writing.

**Experimentation issues**.

- **Lack of baselines** In Sokoban, the work currently compares to one baseline, DRC in Sokoban. DRC is a good choice for a baseline, since it also is a "learning to plan" baseline.

  - However, it is worth comparing the learning to plan strategy against hand-crafted planning and other ways of using the world model for RL. For example, Dyna and model-based value expansion are two alternative ways to leverage a world model for RL. I would like to see competitive baselines from these families of MBRL agents (e.g. DreamerV3 and STEVE).

  - The authors claim to omit common baselines since they want to investigate the benefit of the Augmented MDP. If so, I would like to see the Augmented MDP be applied to a few other RL algorithms (Rainbow, DreamerV3, etc.). Then the focus isn't on raw performance, but rather how much Augmented MDP improves over the vanilla MDP.

- **Low number of seeds (3)** reduces my confidence in the results. Please run at least 5 seeds before reporting metrics, and even better, use IQM and 95% bootstrapped confidence intervals.

- **Dual network ablation:** There is no ablation on the dual network itself. What happens if we just use 1 big RNN with the same amount of parameters to predict all quantities?

- **Parameters for Thinker vs. baselines:** Is it possible the gains from Thinker are just from the increase in world model parameters compared to baselines?


**Poor Writing and Presentation.**

This paper unfortunately suffers heavily from writing and presentation issues, which I believe will make it inaccessible to readers. The authors tend to write dense paragraphs about the important  ideas and mechanisms, rather than summarizing points succinctly and relying on visual figures. It seems like the paper is easy to understand for those deep in the subfields of "learning to plan" and MuZero style of MBRL. However, readers not in this particular area will find it hard to grasp.

L88 - The paragraph on explaining the planning stage is verbose and hard to understand. The provided Figure 2 hides a lot of the details in the planning stage.

I was initially a bit confused between K and L. Thought that K was the max length of 1 rollout. But instead, it seems like K is the # nodes of a planning tree, which is composed of several rollouts all starting from the same root node. Would have been nice to just have a figure of the planning tree to avoid this confusion, perhaps Figure 2 can be updated to have these details?

Some more explanation of planning reward is needed. How does this avoid penalizing searching for paths with low rollout? Is this because we are using Max instead of mean return?



**Questions:**

I think there is a lot of potential in this paper. It needs a good amount of polishing on the writing side to make the idea digestible to a general reader. Next, the experiment section can be improved in a variety of ways - more baselines, ablations, and seeds.

Some additional questions:

Why is the non-stationary network needed? Couldn't we get value and policy distributions directly by evaluating the policy and value functions on the predicted states?

Why does K=1 ablation perform poorly? I thought this would be equivalent to the Raw-MDP baseline.

What is the agent policy vs. the model policy? Is the agent policy the actor in IMPALA and the model policy the world model's predicted action probabilities?

**Limitations:**

The limitations section is already quite fair. However, there are some missing pieces.
1. Computational Cost. MBRL algorithms can be very slow to run in comparison to model free RL.  I would like to see more information about the computational and resource cost of Thinker, and how it compares to the model-free baseline. How many GPUs does it require, how long does it take to run, and what are the main performance bottlenecks?
2. Hyperparameter tuning.  There seems to be quite a bit of hyperparameters.

---

> ### Author Rebuttal · Authors · 2023-08-09
>
> We thank the reviewer for the constructive comments and detailed feedback! We appreciate the reviewer recognizing the potential of our paper while also raising valid concerns regarding the experiments. The weaknesses are addressed as follows:
>
> **Insufficient baselines** - Please refer to (G1) in the global response.
>
> **Number of seeds** - We have added two more seeds to the Sokoban run, bringing the total to five seeds. The results can be found in Fig A4. In general, Thinker’s performance on Sokoban has a very low variance, so having more seeds won’t affect the conclusion.
>
> Regarding Atari, we acknowledge that using more seeds can enhance result reliability. However, the significant costs associated with it (55 runs, with each run requiring 7 days of training on two A100 workstations) prevent us from incorporating a larger number of seeds. As our code is fully public, one can also reproduce our results easily. Moreover, we have not claimed that our results would achieve state-of-the-art performance in Atari, where a more number of seeds might be necessary. Lastly, we kindly ask the reviewer to consider the limited budget we have.
>
> **Dual network ablation** - Please refer to (G2) in the global response.
>
> **Parameters for Thinker vs baselines** - Please refer to (G3) in the global response.
>
> **Poor writing**- We thank the reviewer for pointing out the unclear aspects of the paper, and we will improve the clarity of the paper as follows.
>
> **The paragraph explaining the planning stage** - The paragraph in line 88, along with Figure 2, is intended to give an overview of the planning stage. We acknowledge that it may be too verbose. We will start a new paragraph after *the new real state* on line 96 and remove the line *In each of the K steps, the agent sees....*, which should be better explained in Section 3.2. We also include a better version of Figure 2 that should be clearer, which can be found in Fig A1 of the attached pdf.
>
> **About K and L** - In our paper, K represents the total number of steps taken during a planning stage. Of these steps, K-1 correspond to the 'imaginary actions', which is the same as the number of nodes the agent traverses within a tree. It's worth noting that this doesn't equate to the total number of nodes present in the tree, because the agent can traverse the same node multiple times. On the other hand, L denotes the maximum depth the agent can reach during a rollout. If the agent's search hits this depth (L), it will be reset to the starting point or root node. Typically, the value of L (for instance, L=5) is smaller than K (for instance, K=20).
>
> **Planning reward** - Due to page constraints, we have detailed the planning reward in Section H of the appendix. The reviewer is right: using `max` over `mean` avoids penalizing rollouts with low returns. As elaborated in the appendix, the planning reward acts as a heuristic, guiding the agent to maximize the highest possible rollout return. This gives the agent incentive to try rollouts that have both a low expected return and high variance. If we were to use `mean` instead of `max`, the agent would not have incentives to try rollouts with low expected returns.
>
> The questions raised are addressed as follows:
>
> > Why is the non-stationary network needed?
>
> First, the predicted states may not be perfect, especially during early training, so directly applying a value or policy function to predicted states may lead to inaccurate outputs. As the non-stationary network itself is an RNN, one can recover the MuZero network's way of predicting future policies and values, even when the predicted states are all wrong. Secondly, the RL agent is an RNN that sees both the past imaginary transitions and real transitions. The RL agent receives the augmented state as input in each recurrent step instead of the raw state. Thus, the value and policy function from the RL agent cannot be directly applied to predicted states. Another way of seeing this is that the RL agent's policy acts like a system 2 policy that requires heavy planning, while the model's policy acts like a system 1 policy that only requires a single feedforward pass. The system 1 policy serves as the distilled version of the system 2 policy and guides the search in the model, so they cannot be replaced by one another.
>
> > Why does K=1 ablation perform poorly?
>
> Please refer to our response to Reviewer WUsk (the second question).
>
> > Is the agent policy the actor in IMPALA and the model policy the world model's predicted action probabilities?
>
> Yes.
>
> The limitations raised are addressed as follows:
>
> **Computational Cost**
>
> We acknowledge the need for a more thorough discussion on the computation cost and will add a discussion in the appendix. Please refer to our response to Reviewer WUsk regarding computation cost (the last weakness).
>
> The main bottleneck of the algorithm is training the actor-critic network, and that is why replacing the RNN in the actor-critic network with an MLP can significantly reduce training time. The bottleneck becomes training the model if we use an MLP actor-critic network.
>
> **Hyperparameter tuning**
>
> Yes, our method does introduce additional hyperparameters, specifically the number of planning steps, K, and the maximum search depth, L. However, we found that our selected hyperparameters are largely transferable across different tasks. As evidence, our Atari results were achieved by directly applying the hyperparameters from the Sokoban task, with the only modifications being an increase in network depth and a decrease in the learning rate. The same set of hyperparameters is used on 56 environments. This suggests that extensive hyperparameter tuning is not necessary for most environments.
>
> We hope that the answers provided above adequately address the issues raised by the reviewer, and we kindly ask the reviewer to consider adjusting the score based on our responses. We are grateful for the reviewer’s comments and welcome any further questions!

---

> > ### Comment · Reviewer_9fjR · 2023-08-14
> >
> > Thank you for the clarifications and new experiments.
> >
> > The additional baselines and evaluations clears up my concerns about experimentation.
> >
> > The new figure does explain the Thinker planning process more clearly.
> >
> > I see the point about K, L, and the fact that the tree can reuse nodes if there are repeat states. Could you also add a figure showing some example trees with varying K and L? That would be very easy for readers to quickly get the idea and not get confused from just reading the text.
> >
> > Updating my score to a 6.
> >
> > ---
> >
> > With my immediate concerns cleared up, I would like to add some more discussion.
> >
> > First, my opinion is that Thinker is tightly coupled with its model-based RL algorithm. However, conceptually, it seems like we can use other RL algorithms with the Thinker-augmented MDP. It would be very interesting to compare commonly used RL algorithms with their Thinker counterparts, like SAC, PPO, etc. Perhaps some RL agents will be better with Thinker, and others will not depending on their design choices. At the very least, the authors should include this in the limitations and pose it as a question for future work.
> >
> > Next, planning with K steps every env step seems costly. Can we come up with a version of Thinker that does variable length planning?

---

> > > ### Author Response · Authors · 2023-08-16
> > >
> > > Thank you for your comment and the updated score. In the main paper, we will add a figure showing an example search tree with the corresponding imaginary actions, to better clarify the concepts of planning steps (K), maximum search depth (L), and the reset action.
> > >
> > > > First, my opinion is that Thinker is tightly coupled with its model-based RL algorithm. However, conceptually, it seems like we can use other RL algorithms with the Thinker-augmented MDP.
> > >
> > > Throughout our study, we employed IMPALA—a standard, model-free actor-critic RL algorithm—on the Thinker-augmented MDP. The preference for IMPALA over PPO was mainly due to its superior computational efficiency; IMPALA's design facilitates parallel threads for collecting self-play transitions. We postulate that the performance of PPO or SAC on the Thinker-augmented MDP would align closely with IMPALA's since all these are variants of actor-critic algorithms. We agree that it would be very interesting to experiment with how the Thinker-augmented MDP interacts with other model-free RL algorithms, especially value-based ones like Q-Learning. As cited in line 333 of the paper, *"Exploring how other RL algorithms perform within the Thinker-augmented MDP is an additional direction for future work."*
> > >
> > > > Next, planning with K steps every env step seems costly. Can we come up with a version of Thinker that does variable length planning?
> > >
> > > We have briefly experimented with introducing an extra action in the augmented MDP, so the RL agent can choose to stop planning and act. We found that the final performance at 5e7 frames is similar, and the number of planning steps is reduced by around half. We did not include this in the paper as (i) the focus of the paper is the performance of Thinker given a limited number of frames instead of wall time, (ii) as Thinker opens up a new way of using a learned model in model-based RL, we believe the algorithm should be as simple as possible so as to allow future work to build on it easily.

---

### Official Review · Reviewer_U2R9 · 2023-07-19

**Soundness:** 4 excellent
**Presentation:** 3 good
**Contribution:** 3 good
**Rating:** 7
**Confidence:** 3

**Summary:**

- This paper introduces a novel algorithm, Thinker, which learns how to plan by interacting with a learned world model using “imaginary actions”—negating the need for hand-designing a planner. The algorithm produces SOTA results on Sokoban, and its efficacy on Atari 2600 is demonstrated as well.

**Strengths:**

- The algorithm design is natural given the motivation of the work.
- The introduced algorithm has a number of novel components, including the planning stage used over the augmented MDP, and the augmented reward, and the duel network architecture.
- Thinker produces state-of-the-art results on Sokoban (over previous state-of-the-art, DRC), and in Atari 2600, the Thinker-augmented approach outperforms the non-Thinker-augmented baseline.
- The writing is clear and relatively concise, the paper is well-structured, the significance of the work is well-communicated and compelling, and the related work section appears to be complete.

**Weaknesses:**

- The authors note “as the goal of the [Atari 2600] experiments is to investigate the benefits of the augmented MDP, we do not include other common baselines here” (line 307). While this point is understood, the results on Atari would have been more compelling if Thinker were to have outperformed comparable learned planning approaches, e.g. those mentioned in related work.
- Figure 1 is a bit difficult to read—it is hard to note the difference between each of the successive images. Better visualization to highlight the differentials (and zooming in on the level, maybe) could have been employed.

**Questions:**

- In line 234, various approaches using gradient updates to plan within a planning stage are cited. These approaches do not appear to be used in evaluation as baselines/ablations—why not?
- Could some of the improved results be due to the novel duel network architecture, rather than the new algorithm? I don’t see an ablation for this in the main body or supplementary material.
- How were the output statistics chosen? Could alternatives have been used?
- I am confused by the phrase “the real action at the first K-1 steps” (132), since I understand that the fist K-1 actions are imaginary. Is it possible the authors are trying to say that we’re not using real actions for the first K-1 steps? It seems like these real actions are not simply “not used”, but they don’t exist in the first place. The structure of the sentence also makes it unclear to me which “imaginary action” we are talking about (I would suggest maybe ditch the last comma for clarity, since this also corresponds to the last step, K).

**Limitations:**

- The authors note the limitations that (i) the algorithm carries a large computational cost, (ii) only rigid planning is supported, i.e. requiring agent to roll out from the root state and restricts it to planning for fixed number of steps, and (iii) a deterministic environment is assumed. I agree with this authors that these elements are appropriate for future work.
- No other major limitations stand out to me.

---

> ### Author Rebuttal · Authors · 2023-08-09
>
> We thank the reviewers for the constructive comments and detailed feedback! We appreciate the reviewer’s recognition of our work. The weaknesses are addressed as follows:
>
> **Insufficient baselines** - Please refer to (G1) in the global response.
>
> **Clarity of Figure 1** - We will use a bordered window to emphasize the agent and enlarge the images. Additionally, we will update the border color of the rollout image to match the rollout summary on the left.
>
> The questions raised are addressed as follows:
>
> > In line 234, various approaches using gradient updates to plan within a planning stage are cited. These approaches do not appear to be used in evaluation as baselines/ablations—why not?
>
> This is because [1, 2] requires perfect information about the MDP, meaning one needs a perfect simulator of the environment, while [3, 4] requires a continuous action space. We focus on the conventional RL setting, where prior knowledge about the MDP is not available, and only consider MDPs with a discrete action space.
>
> > Could some of the improved results be due to the novel duel network architecture, rather than the new algorithm? I don’t see an ablation for this in the main body or supplementary material.
>
> Please refer to (W3) in the global response for the ablation results on imaginary actions. If we replace the learned imaginary action with a random imaginary action, performance drops significantly. As such, the improved results are not only due to the novel dual network architecture but also the learning of imaginary actions in the augmented MDP. Section G in the appendix contains additional comparisons with MCTS.
>
> > How were the output statistics chosen? Could alternatives have been used?
>
> The most important output statistics are values and policies, inspired by the following rationale: Values allow evaluating a rollout without the need to roll till the end of the episode, while policies allow the search to be more efficient by providing hints to the agents. Both of these quantities are critical - see Fig F4 in the appendix, which shows learning cannot occur without these two quantities. We do not know if there are any alternatives that can replace these two quantities.
>
> > It seems like these real actions are not simply “not used”, but they don’t exist in the first place.
>
> Yes, one can interpret that the real actions at the first K-1 planning steps, or the imaginary actions at the K planning step, do not exist at all. However, since the actor-critic network outputs both real action and imaginary action with different heads on each planning step, so in terms of the code, those actions do exist (but they will be discarded). We acknowledge that it is clearer if we state the first K-1 actions are imaginary actions while the K action is a real action instead of saying the other actions are not used. We will revise the paper to reflect this. Thank you for the suggestion.
>
> We hope that the answers provided above adequately address the issues raised by the reviewer. We are grateful for the reviewer’s comments and welcome any further questions!
>
> **References**
>
> [1] Thomas Anthony, Robert Nishihara, Philipp Moritz, Tim Salimans, and John Schulman. Policy gradient search: Online planning and expert iteration without search trees. arXiv preprint arXiv:1904.03646, 2019.
>
> [2] Arnaud Fickinger, Hengyuan Hu, Brandon Amos, Stuart Russell, and Noam Brown. Scalable online planning via reinforcement learning fine-tuning. Advances in Neural Information Processing Systems, 34:16951–16963, 2021.
>
> [3] Aravind Srinivas, Allan Jabri, Pieter Abbeel, Sergey Levine, and Chelsea Finn. Universal planning networks: Learning generalizable representations for visuomotor control. In International Conference on Machine Learning, pages 4732–4741. PMLR, 2018.
>
> [4] Mikael Henaff, William F Whitney, and Yann LeCun. Model-based planning with discrete and continuous actions. arXiv preprint arXiv:1705.07177, 2017.

---

> > ### Comment · Reviewer_U2R9 · 2023-08-16
> >
> > The authors have addressed my questions adequately and I maintain my positive opinion of the paper.

---

### Official Review · Reviewer_sy9V · 2023-07-19

**Soundness:** 1 poor
**Presentation:** 2 fair
**Contribution:** 1 poor
**Rating:** 5
**Confidence:** 3

**Summary:**

This paper presents a model-based RL approach, Thinker, that learns a world model and plans to take actions by generating imaginary rollouts with the learned world model. The paper claims that Thinker is a general method for any RL algorithms and does not rely on any hand-crafted planning algorithm. Experiments were conducted on Sokoban and the Atari 2600 benchmark, showing better performance over model-free RL methods.

**Strengths:**

1. The basic idea of learning world models and training model-based RL policies with the learned models is technically solid. Experiments indeed show better performance against model-free baselines

2. The method description is clear and it is good to see performance in different domains.

**Weaknesses:**

1. I am very surprised that Thinker is not compared against model-based RL baselines.

2. The contribution is unclear to me:

    a) I do not understand the claim about not having a hand-crafted planning algorithm. Based on the method described in the paper, you did introduce an MCTS-type of planning algorithm. This claim needs to be explained and justified.

    b) What is the difference between your method and other model-based RL methods? Particularly, it looks similar to MuZero, except that you have a different network architecture for learning the world model.


**Questions:**

Please address my questions in the review.

**Limitations:**

It is hard for me to judge whether limitations have been sufficiently addressed since I do not even understand what new technical advances have been proposed here.

---

> ### Author Rebuttal · Authors · 2023-08-09
>
> We thank the reviewer for the questions. The questions raised are addressed as follows:
>
> **Insufficient baselines** - Please refer to (G1) in the global response.
>
> **The difference with MuZero** - Please refer to the first three paragraphs of the global response and reviewer #fWg3’s summary of the algorithm. In particular, *we did not introduce any MCTS in our algorithm*. An actor-critic is trained to select imaginary action (instead of the UCB formula in MCTS) and real action (instead of based on visit counts in MCTS) by maximizing the return in the augmented MDP. An RL agent can learn a variety of planning algorithms depending on its needs. Please refer to Appendix G for the analysis of the agents’ learned planning algorithm, which differs significantly from hand-crafted algorithms such as MCTS.

---

> > ### Comment · Reviewer_sy9V · 2023-08-16
> > **Thanks for your response**
> >
> > After reading the responses, I understand better about the contribution. I also appreciate the additional results. I have increased my rating accordingly.

---

### Official Review · Reviewer_aBLR · 2023-07-24

**Soundness:** 2 fair
**Presentation:** 2 fair
**Contribution:** 2 fair
**Rating:** 5
**Confidence:** 3

**Summary:**

This work presents Thinker, a new model-based reinforcement learning algorithm that achieves state-of-the-art performance on the puzzle video game, Skoban. The main contribution made by the authors is using a dual network architecture which addresses the sample inefficiency problem that other algorithms such as MuZero have. This dual-network setup consists of two sub-networks, namely a stationary and non-stationary network.  The stationary network takes as input the current state and a sequence of raw actions and predicts future states, rewards and episode termination probabilities. The non-stationary network takes as input both the stationary network’s inputs and its predicted next states.

In addition to the supervised training loss over the four predicted quantities, the authors proposed another L2 loss that encourages the encoded representation of the stationary network’s predicted state to match the encoded representation of the true state observed. The encoder helps the non-stationary network focus on only encoding task-specific information. The static network encodes static policy-independent information, while the non-static network encodes information that changes as the policy changes. Lastly, the encoder is only updated using the loss of the non-static network. The static network uses this encoder, without updating it, and updates its future state predictions to minimise the L2 squared distances in latent space.


**Strengths:**

The authors show that their proposed system achieves a new state-of-the-art result on Sokoban. Furthermore, they proposed using an L2 distance in a latent representation instead of using the full state. This makes the entire system much more scalable. They further provide a detailed breakdown of the algorithm in the appendices and open source their code.

**Weaknesses:**

The main weakness I see is that the use of these dual networks is not well motivated. Along with a better motivation an additional experiment or two is needed to demonstrate that they outperform a single network setup.

Furthermore, it would be helpful to directly compare against open-source versions of models such as DreamerV3 and MuZero. This can help motivate in what ways their approach is superior.


**Questions:**

1. On line 245 you state, “ In contrast to these methods, our proposed dual network fits the next-state prediction with a feature loss that enables visualization and prioritizes the learning of task-relevant features.” How does a feature loss enable visualisation? Especially if the agent only needs to remember task-relevant information. When the stationary network predicts the next state, would it not be able to ignore parts of the state that the encoder does not use? Therefore the latent space distance is still close to zero, but the predicted state might have some missing information.


2. On line 210 you state, “It is important to note that g is not being optimized when minimizing this loss, and ŝt+l does not receive gradients from the non-stationary network, as the two sub-networks are separately optimized. Further details regarding the model’s training can be found in Appendix B.” If it uses the encoder from the non-stationary network would that not make the stationary network’s updates non-stationary as well? As the policy changes the encoder (g)’s parameters might change. Which in turn changes the next state representation that gets generated.


3. On line 192 you state, “Nonetheless, this approach suffers from sample inefficiency since it discards information from the future state, which carries a rich supervised learning signal.” In what way does the dual network approach address the inefficiency problem that you state other methods such as MuZero and DreamerV3 have?


**Limitations:**

This proposed solution currently only works for environments with deterministic dynamics. Furthermore, the planning component is quite computationally expensive, which might make it too expensive to use in many scenarios.

The main limitation I see is that there has not been enough evidence provided that the dual network setup performs better than alternatives. Especially since the stationary network actually uses components from the non-stationary setup. It would be of great benefit to have additional experiments that show the superiority of this architecture when compared to alternatives (such as predicting every value using a single network).

---

> ### Author Rebuttal · Authors · 2023-08-09
>
> We thank the reviewers for the constructive comments and detailed feedback! The summary gives a clear description of the proposed dual network, but we would like to point out that the main focus of the paper is the augmented MDP and learning to plan (thus the paper’s title), and dual network is only a part of the overall algorithm. The weaknesses are addressed as follows:
>
> **Insufficient baselines** - Please refer to (G1) in the global response.
>
> **Ablation on the dual network** - Please refer to (G2) in the global response.
>
> The questions raised are addressed as follows:
>
> > How does a feature loss enable visualisation?
>
> Previous work [1] has shown that autoencoder training can rely solely on a feature (or perceptual) loss, omitting pixel-space losses, while still achieving good pixel-level reconstruction. We conjecture the reasons for that are (i) feature loss retains the spatial structure as convolutional layers are used, (ii) usually activations from early layers (1-5) are employed so the features are still close to the image itself.
>
> It is common in computer vision that the network used to compute the feature (or perceptual) loss is pre-trained and fixed. However, Thinker deviates from this norm by utilizing features derived from a network that is concurrently trained to predict policy and values. Nevertheless, we believe the intuition is the same. Moreover, arguably because the feature loss should be satisfied in the dynamic of a concurrently trained network, it incentivizes the network to also predict well in the image space. Finally, it’s worth noting the domain of games is much simpler than the domain of natural images, possibly another reason why feature loss reconstructs images so well in our experiments.
>
> We agree that the feature loss does not necessarily lead to accurate reconstruction. Indeed one can see some artefacts in the video on the main paper, especially in Atari games. We believe these artefacts are typical for feature loss. For instance, Fig 6 in [1] shows that by using L2 loss, the network omits the purple ball in the reconstructed image, but the feature loss reconstructs it with a dotted grid-like object. The same dotted grid-like object can be seen at the 30:44 timestamp of the video in the paper. Nonetheless, we observe that reconstructed images from the video are of high quality and can be easily interpretable.
>
> > If it uses the encoder from the non-stationary network would that not make the stationary network’s updates non-stationary as well?
>
> Yes, the loss function is non-stationary, which changes as the non-stationary network is updated. But we did not find any issues with the non-stationary loss. An ablation analysis of directly using L2 loss on state space is also included in Fig A3, showing that the non-stationary loss yields better performance than the vanilla L2 loss on the raw state space.
>
> We agree that the name *stationary network* may be confusing as it uses a non-stationary loss. As such, we will revise the naming from *stationary network* to *state-reward network* and *non-stationary network* to *value-policy network* to better reflect their nature.
>
> > In what way does the dual network approach address the inefficiency problem that you state other methods such as MuZero and DreamerV3 have?
>
> We include the feature loss such that the learned model has to predict future features of the environment. This provides an additional supervised learning signal for the model. The stationary network (or the state-reward network) is designed to predict future states, which then act as intermediary steps or hints for estimating future output statistics. Take, for instance, the task of predicting the last reward after making the moves (up, up, up, right) in Sokoban. Predicting the reward becomes easier based on projected future frames. For example, if the character is anticipated to be on the left of a box after three upward moves, then the likely reward prediction would be one. The primary role of the stationary network (or the state-reward network) is to make these future frame predictions. While, in theory, a neural network could autonomously learn this intermediary step, doing so would necessitate significantly more data.
>
> As seen in Figure A2, MuZero generally requires a much larger amount of frames to reach a competitive performance. As for DreamerV3, we believe its model is data efficient (it performs well in the Atari 100k benchmark) as it also includes image reconstruction loss in training the model. However, DreamerV3 still performs a lot worse than Thinker in Sokoban - this is likely due to the extensive planning imposed on the Thinker algorithm. Again, Thinker differs from both MuZero and DreamerV3 in both how a model is used (augmented MDP in Thinker v.s. MCTS in MuZero v.s. Dyna-like simulation in DreamerV3) and the model itself (dual network in Thinker v.s. single network in MuZero v.s. VAE in DreamerV3, which does not predict future values or policies that are necessary for an agent to learn in the augmented MDP).
>
> We hope that the answers provided above adequately address the issues raised by the reviewer, and we kindly ask the reviewer to consider adjusting the score based on our responses. We are grateful for the reviewer’s comments and welcome any further questions!
>
> **References**
>
> [1] Gustav Grund Pihlgren, Fredrik Sandin, Marcus Liwicki. Improving Image Autoencoder Embeddings with Perceptual Loss, arXiv preprint arXiv:2001.03444, 2020

---

> > ### Comment · Reviewer_aBLR · 2023-08-15
> >
> > I want to thank the authors for their response and for the proposed updates to the paper. I think the updated network names are much better. Furthermore it is now easier to verify the necessity of using dual networks over just a single network. The results also seem to indicate that Thinker significantly outperforms Dreamer-v3 and MuZero on Sokoban when using environment frames as a reference.
> >
> > I have updated my score to 5.
> >
> > One question I still have is how Tinker compares against the other algorithms when considering wall clock time. Do the other algorithms perform better using this metric? If they perform better here, what is the main advantage of Tinker? Would it maybe perform better in environments with computationally intensive/slow environment steps?

---

> > > ### Author Response · Authors · 2023-08-16
> > >
> > > Thank you for your comment and the updated score. Thinker does indeed outperform both Dreamer-v3 and MuZero on Sokoban when using environment frames as a reference. While Dreamer-v3 isn't specifically tailored for the planning domain, its performance understandably lags behind RL algorithms designed for planning. Regarding MuZero, its model's sample inefficiency means it needs considerably more frames than Thinker (it requires 20 billion frames for Atari). Moreover, substituting MCTS with an RL agent introduces greater flexibility in both planning and acting. For example, if planning proves non-essential, the RL agent can bypass the search and behave like a typical plan-free RL agent; or, as Appendix G demonstrates, an RL agent might choose to stick to an existing plan unless encountering an unfavorable leaf node, which is not possible with MCTS.
> > >
> > > When we take wall time as a reference, the same conclusion holds -  Thinker outperforms both Dreamer-v3 and MuZero on Sokoban when using wall time as a reference. As we mentioned in our response to Reviewer WUsk, Dreamer-v3 and MuZero require similar or longer training times.
> > >
> > > Wall time is not the primary focus of our paper, and various modifications and implementation details can enhance Thinker's wall time. For instance, substituting the RNN in the actor-critic with an MLP and reducing the planning steps from 20 to 10 could decrease the training time by over 70% while maintaining similar performance levels. Our interest mainly lies in evaluating the performance of an RL algorithm based on a fixed number of frames, a more typical reference point as seen in the Atari-200m benchmark.

---

### Official Review · Reviewer_WUsk · 2023-07-26

**Soundness:** 3 good
**Presentation:** 4 excellent
**Contribution:** 3 good
**Rating:** 6
**Confidence:** 3

**Summary:**

This paper presents the Thinker algorithm, a method that augments an MDP such that an agent takes 'imaginary actions' in a learned model prior to executing actions in the environment. This allows the agent to incorporate planning strategies into a policy learned with RL. State-of-the-art performance is achieved on Sokoban and competitive results are shown on Atari.

**Strengths:**

- The paper is well-written and polished.
- The appendix is thorough and documents implementation details and architecture details. Source code is provided for reproducibility.
- The proposed algorithm for augmenting an MDP to induce planning behavior in an agent seems to be original, as is the architecture of the dual network for learning the model. The results on Sokoban and certain Atari tasks also appear to be quite strong.
- An analysis of the learned behavior of agents (e.g., when resetting happens during planning, what types of imaginary and real actions are taken) is provided in the Appendix. This is useful for characterizing the behaviors of the learned policies.

**Weaknesses:**

- Comparison is given on Sokoban to DRC and to Rainbow on Atari. Most significantly, I think this work would benefit from comparison to model-based RL baselines to help contextualize how well the Thinker algorithm performs.
- The method has multiple moving parts and several involved design choices (e.g., what elements to include in the augmented state, non-stationary planning reward, dual network architecture), though ablations are provided for some of these choices in Appendix F.
- Related to the above point, regarding the dual network. The paper notes that "one could adopt the same architecture and training methodology used by MuZero" but that "[MuZero] suffers from sample inefficiency since it discards information from the future state." It would be useful to provide a quantitative comparison/ablation here, where the encoder/unrolling function/prediction function are learned in the manner of MuZero, as described, in order to justify the proposed architecture.
- The augmented MDP seems to increase the complexity of the MDP (in the state-action space as well as the time-horizon). An expanded discussion on the computational cost of both training and evaluation, compared to using the raw MDP and also to other model-based methods, would be useful.

**Questions:**

I would appreciate clarification on the items described in weaknesses; some additional questions:
- Is it correct that the augmentation increases the cardinality of the action space from |A| to 2|A|^2? Does this adversely affect the difficulty of learning a policy?
- Is there an intuition for why the K=1 case does worse than in the raw MDP in Figure 5?
- An ablation on the unroll length L is provided in Figure F.3 in the Appendix. Have the authors tested additional values <5 or >10 -- i.e., is there a clear trend here? I am also curious how sensitive the values of L and K are to a particular task.

**Limitations:**

- The authors have noted some limitations in Section 6, including computational cost and rigid planning for a fixed number of steps, including determinism of the MDP.

---

> ### Author Rebuttal · Authors · 2023-08-09
>
> We thank the reviewers for the constructive comments and detailed feedback! The weaknesses are addressed as follows:
>
> **Insufficient baselines** - Please refer to (G1) in the global response.
>
> **Multiple moving parts** - Please refer to (G2) and (G3) in the global response for more ablation analysis.
>
> **Ablation on the dual network** - Please refer to (G2)  in the global response.
>
> **Computation cost** - We acknowledge the need for a more thorough discussion on the computation cost, and will add the following discussion in the appendix when discussing the new baselines:
>
> *As discussed in the main paper, Thinker augments the MDP by adding K-1 steps for every step in the underlying MDP. Consequently, there is an increased computational cost due to the extra K-1 imaginary steps, in addition to the added computational cost of training the model. On our workstation equipped with two A100s, the training durations for the raw MDP, DRC, and Thinker on Sokoban are around 1, 2, and 7 days, respectively.*
>
> *However, when benchmarked against other model-based RL baselines, Thinker's training time is similar or even shorter. For instance, Dreamer-v3 (the open-sourced implementation) requires around 7 days for a single Sokoban run on our hardware, primarily due to its high replay ratio. MuZero, when conducting a standard number of simulations (e.g., 100), requires 10 days of training. It is noteworthy that despite the Thinker's additional training overhead for RL agents, the algorithm compensates by requiring significantly fewer simulations. Furthermore, if one replaces the RNN in the actor-critic network with an MLP (as detailed in Appendix F), without compromising the performance, the computational time for a single Thinker run can be reduced to 3 days.*
>
> The questions raised are addressed as follows:
>
> >Is it correct that the augmentation increases the cardinality of the action space from |A| to 2|A|^2? Does this adversely affect the difficulty of learning a policy?
>
> The cardinality of the action space is 2|A|, as we have (real action / imaginary action, reset) as the new action set. Note that the real actions on imaginary steps (the first K-1 step in a planning stage) and the imaginary actions on a real step (the K step in a planning stage) are not used (or as Reviewer U2R9 points out, they do not exist), so they can be considered to be the same type of action. As the new action space’s cardinality is still linear in |A|, this shouldn’t adversely affect the difficulty of learning a policy.
>
> >Is there an intuition for why the K=1 case does worse than in the raw MDP in Figure 5?
>
> The reason for this is that, instead of providing the raw image as input to the actor-critic network for all K steps, we feed in the model's hidden state. While this hidden state might be more effective for predicting future states and rewards of the MDP, it can be more difficult for learning a good policy. This is particularly the case because the model undergoes updates throughout the training process. This approach potentially explains why in a small number of Atari games, the actor-critic applied to the raw MDP occasionally outperforms the one applied to the Thinker-augmented MDP, as highlighted in Appendix E3:
>
> *Interestingly, the actor-critic applied to the raw MDP occasionally outperforms the one applied to the Thinker-augmented MDP. This could be attributed to the fact that we provide the agent with the model’s hidden state instead of the true or predicted frames. It is conceivable that the actual frame is simpler to learn from, given that the model learns the encoding of frames by minimizing supervised learning loss rather than maximizing the agent’s returns.*
>
> Future research could consider providing both the raw image and the model's hidden state as inputs to the RL agent to potentially address this issue.
>
> >Have the authors tested additional values <5 or >10 -- i.e., is there a clear trend here?
>
> We have not conducted tests for scenarios where L < 5. However, we have tested the case of L=20 (implying no restriction on search depth given that the number of planning steps, K, is 20) and conducted a brief behavioural analysis. Notably, the learned maximum search depth within a planning stage averages between 16 and 17. This suggests that RL agents typically prefer to unroll a single, extended rollout. In particular, for some simpler Sokoban levels, the RL agent can imagine solving the level in the first real step. Nevertheless, the initial learning speed is notably slower, while the final performance observed is marginally worse. Our conjecture for this trend is that longer training is required for the model to learn to unroll over a span of 20 steps. We did not include this result in the paper as the results are from the early version of Thinker, where the MDP was augmented in a slightly different manner.
>
> We hope that the answers provided above adequately address the issues raised by the reviewer, and we kindly ask the reviewer to consider adjusting the score based on our responses. We are grateful for the reviewer’s comments and welcome any further questions!

---

> > ### Comment · Reviewer_WUsk · 2023-08-12
> > **Reply to Rebuttal**
> >
> > Thank you for the response. I think that the additional ablations (including on the dual network, since it is considered a significant contribution) as well as the additional model-based baselines are necessary additions to the paper, and am glad the authors have included these in their rebuttal. I continue to lean towards accepting the paper.

---

### Author Rebuttal · Authors · 2023-08-09

We thank the reviewers for the constructive comments and detailed feedback. We appreciate that most reviewers recognize the novelty and value of our proposed algorithm, Thinker. We note that some reviewers seem to underestimate the novelty, viewing Thinker as merely a variant of MuZero, or focusing overly much on the architecture as opposed to the core novelty of our approach, which is to unify planning and acting within a single augmented MDP (hence the paper’s title). For reference, Reviewer #fWg3 provided a succinct summary of the algorithm, and a clearer illustration of the algorithm (replacing Fig 2 in the paper) is shown in Fig A1 of the attached pdf.

To clarify the novelty with respect to MuZero,  MuZero can be understood as (i) MCTS and (ii) a learned model, while Thinker can be understood as (i) an RL agent in an augmented MDP and (ii) a learned model with the dual network architecture. Thus, *Thinker does not employ MCTS in any capacity.* While the specific architecture of the model is not our central concern, we do note that the use of a dual network enhances performance, and consider it a significant contribution, just not the main one.

**Thinker is the first work showing that an RL agent can learn to plan with a learned world model in complex environments.** The area of 'learning to plan' is still nascent, largely due to its intrinsic challenges. Prior works either do not satisfy (i) learning to plan, i.e., training a neural network to select imaginary actions, (ii) using a learned model or (iii) being evaluated in a complex environment. IBP [20] shares similar ideas with Thinker, but their models do not predict values and policies, which we found necessary in complex domains (see Fig F4 in the appendix). As such, their work has only been shown to work in simple domains, and (iii) is not satisfied. TreeQN [14] and I2A [15] do not satisfy (i). VIN [19] and DRC [10] do not satisfy (ii). Another line of work [21-24] corresponds to algorithms that perform gradient updates within a planning stage. This requires either a perfect model [21, 24] or simple environments [23], so either (ii) or (iii) is not satisfied. ([22] is an imitation learning algorithm instead of an RL algorithm.)

Common issues raised by reviewers are addressed as follows:

**(G1) Additional Baselines**

We added six new baselines in the Sokoban domain, including Dreamer-v3 [52], MuZero [1], I2A [15], ATreeC [14], VIN [19], and IMPALA with ResNet [35]. We use open-source implementation for Dreamer-v3, with the result being averaged over three seeds. MuZero’s result is taken from [53]. Results of DRC (original paper), I2A, ATreeC, VIN and IMPALA with Resnet are taken from [10]. The results are shown in Fig A2 and Table A1 in the attached pdf. In hard planning domains, planning-orientated RL algorithms, such as DRC, can generally achieve a much better performance than other RL algorithms. And within planning-orientated RL algorithms, DRC usually outperforms others by a wide margin [10]. As such, we only included DRC as the baseline in our submission, but we agree with the reviewers that more baselines would enrich our paper. As for Atari, we added the new baselines of UNREAL [54] and LASER [55] from previous papers, and the results are shown in Table A2. Readers who are interested in more baselines can refer to other papers, as Atari 200m is a common benchmark. Note that the goal of the Atari experiments is to evaluate the benefits of the Thinker-augmented MDP in other domains beyond planning domains, instead of competing with baselines.

**(G2) Ablation on the Dual Network**

We conducted three additional ablation experiments:

(i) Single network that predicts the relevant quantities but not future states, which is the same as MuZero's model.

(ii) Same as (i), but with the addition of predicting states using L2 loss (though the reviewers did not request this run, we believe it provides valuable insights. The experiment is still running, and we expect to complete the training by Aug 13).

(iii) Dual network with L2 state prediction loss (instead of feature loss).

The result is shown in Fig A3.

**(G3) Ablation on Imaginary Action**

To address reviewers' concerns that the performance may be due to the model's increased parameter count or additional features, we included an ablation experiment by using random imaginary actions while training the RL agent only on real actions. The result is presented in Fig A3. We also note that the IMPALA with ResNet shown in Fig A2 has 52.5m parameters, whereas Thinker has 8.5m parameters, so increasing the parameter count does not necessarily improve performance.

All aforementioned figures will be incorporated into the revised appendix. With the new experiments included, *our paper now encompasses over 70 distinct runs, spanning evaluations across 56 environments and 16 ablation studies, in addition to the extensive analysis in the appendix.* Most related works on learning to plan have fewer experiments. For example, [24], published in NeurIPS 2021, evaluated their algorithm on two environments with a single baseline. We hope that our new experiments, along with consideration of the computational cost, could address the reviewers’ concerns regarding experiments.

**References**

[1-51] See the main paper’s references.

[52] Hafner, D., Pasukonis, J., Ba, J., & Lillicrap, T. (2023). Mastering diverse domains through world models. arXiv preprint arXiv:2301.04104.

[53] Hamrick, J. B., Friesen, A. L., Behbahani, F., Guez, A., Viola, F., Witherspoon, S., ... & Weber, T. (2020). On the role of planning in model-based deep reinforcement learning. arXiv preprint arXiv:2011.04021.

[54] Jaderberg, Max, et al. "Reinforcement learning with unsupervised auxiliary tasks." arXiv preprint arXiv:1611.05397 (2016).

[55] Schmitt, S., Hessel, M., & Simonyan, K. (2020). Off-policy actor-critic with shared experience replay. In ICML (pp. 8545-8554). PMLR.

---

### Decision · Program_Chairs · 2023-09-21

**Decision:**

Accept (poster)

**Comment:**

After introducing the additional baselines, the reviewers unanimously recommended the acceptance of the paper.
The authors should explicitly compare to model-based approaches, as was done in the rebuttal, and relate to the algorithms mentioned in the rebuttal, including [56].

[56] https://arxiv.org/abs/2202.01108